# FT-AED: Benchmark Dataset for Early Freeway Traffic Anomalous Event Detection

**Austin Coursey**[1,2,†]**, Junyi Ji**[1,3,†]**, Marcos Quinones-Grueiro**[1]**, William Barbour**[1]**,**
**Yuhang Zhang**[1,3]**, Tyler Derr**[2]**, Gautam Biswas**[1,2]**, Daniel B. Work**[1,2,3]
[1]Institute for Software Integrated Systems, [2] Department of Computer Science
[3]Department of Civil and Environmental Engineering
Vanderbilt University
[†]Corresponding authors: {austin.c.coursey, junyi.ji}@vanderbilt.edu

## Abstract

Early and accurate detection of anomalous events on the freeway, such as accidents, can improve emergency response and clearance. However, existing delays and mistakes from manual crash reporting records make it a difficult problem to solve. Current large-scale freeway traffic datasets are not designed for anomaly detection and ignore these challenges. In this paper, we introduce the first large-scale lane-level freeway traffic dataset for anomaly detection. Our dataset consists of a month of weekday radar detection sensor data collected in 4 lanes along an 18-mile stretch of Interstate 24 heading toward Nashville, TN, comprising over 3.7 million sensor measurements. We also collect official crash reports from the Tennessee Department of Transportation Traffic Management Center and manually label all other potential anomalies in the dataset. To show the potential for our dataset to be used in future machine learning and traffic research, we benchmark numerous deep learning anomaly detection models on our dataset. We find that unsupervised graph neural network autoencoders are a promising solution for this problem and that ignoring spatial relationships leads to decreased performance. We demonstrate that our methods can reduce reporting delays by over 10 minutes on average while detecting 75% of crashes. Our dataset and all preprocessing code needed to get started are publicly released at `https://vu.edu/ft-aed/` to facilitate future research.

## 1 Introduction

One primary concern of a freeway traffic management center revolves around anomalous incident detection and response [8]. On the freeway, these events could be vehicle accidents, vehicle malfunctions, slow-moving vehicles, or severe weather conditions. Early and accurate detection could reduce the risk of secondary accidents and ease traffic congestion [6]. A current operational system for handling incidents on highways involves staff continuously monitoring live camera feeds. They must wait to receive an accident report before initiating management actions, and these actions are only triggered after the incident is either reported by a driver or observed by an officer through the cameras. These systems highly rely on manual efforts, and there is a clear need to automate the incident detection process to improve intelligent transportation systems [21]. Additionally, numerous incidents go unreported but lead to more severe secondary crashes as traffic conditions worsen [11].

Recently, researchers have explored deep-learning techniques for incident detection. Some of these have been supervised, using common approaches like Convolutional Neural Networks [6], Support Vector Machines and Probabilistic Neural Networks [22], Recurrent Neural Networks [27], and models that combine spatial and temporal information [13, 12]. Accurate crash or incident

38th Conference on Neural Information Processing Systems (NeurIPS 2024) Track on Datasets and Benchmarks.

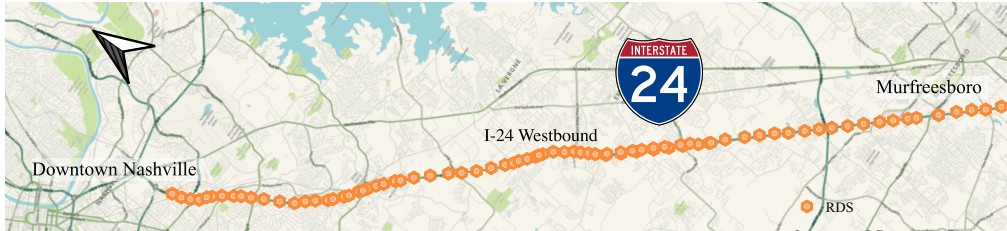

Figure 1: Map of the Radar Detection Systems (RDS) sensor network deployed for data collection. 49 RDS sensors are deployed along Interstate 24 toward Nashville, Tennessee. Each sensor captures speed, occupancy, and volume data for each of the four lanes every 30 seconds.

information may not be available due to reporting delays. Therefore, unsupervised methods for crash detection may be preferred. One way of doing this is by framing the incident detection problem as an anomaly detection problem [30, 5]. As with all deep-learning methods, these approaches rely on the availability of large, realistic freeway traffic datasets.

While large datasets exist for the widely-studied problem of traffic forecasting [18, 7], there is limited data focused on freeway anomalous event detection (see Section 2). Specifically, no current datasets address the fact that, **in the real world, there is a delay in incident reporting**. Imagine that a crash occurs on the freeway. Once it is safe, a witness may notify emergency services. Then, emergency services may dispatch first responders and contact the local traffic management center. Once the traffic management center has assessed the situation and made the appropriate decisions, they may manually log the crash, potentially minutes after it occurred. This challenge necessitates unique metrics and solutions as there is a level of uncertainty and distrust in labeled incident reports.

In this work, we make the following contributions.

1. To the best of our knowledge, we release the **first large-scale lane-level freeway traffic dataset designed for anomaly detection**. This dataset encapsulates traffic states for every workday in October 2023, captured at 30-second intervals using 49 radar detection sensors placed along the Interstate 24 (I-24) corridor, stretching from Murfreesboro to downtown Nashville, Tennessee. Additionally, we sourced true incident labels from the Tennessee Department of Transportation Traffic Management Center and provide anomaly labels based on expert analysis of the traffic speed profiles. This dataset is publicly released along with code at `https://vu.edu/ft-aed`.
2. We define a **new problem** of lane-level anomaly detection, emphasizing the challenges of delayed crash reporting.
3. We establish the Freeway Traffic Anomalous Event Detection (FT-AED) benchmark according to our collected data and problem definition. We conduct a **thorough benchmarking of various autoencoder-based deep-learning anomaly detection methods** to establish a baseline for the proposed novel task of lane-level freeway anomaly detection.

## 2 Background

### 2.1 Existing Datasets

Many datasets for traffic anomaly detection focus on detecting anomalies from videos [24]. Recent ones consist of CCTV videos of accidents in India [26], dash cam accident videos [31], and college campus security camera videos [17]. While videos are rich in information, they are computationally expensive to store and process. Obtaining high-quality anomaly annotations can be a labor-intensive process. Additionally, videos may raise privacy concerns. For these reasons, deploying vision-based anomaly detection models along major freeways may not always be feasible. Therefore, large-scale freeway traffic datasets typically use sensor measurements.

A number of large sensor-based freeway traffic datasets exist, and these have been used extensively for tasks like traffic forecasting [7]. One of the most popular sources of freeway sensor data is the Performance Measurement System (PeMS) [1]. This data source comprises estimated traffic measurements from thousands of loop detectors across California freeways. Large subsets of this data source have been processed and publicly released, the most recent of which contains billions of data samples [18]. Two popular smaller subsets of PeMS are PEMS-BAY and METR-LA [14].

Table 1: High-level comparison of datasets that have been used for freeway anomaly detection. "Anomalies" means whether anomaly labels are included with the dataset. With the METR-LA and PEMS-BAY datasets, anomalies can be externally sourced but are not included by default. Our dataset contains lane-level features captured every 30 seconds from radar sensors. This increased granularity allows for quicker incident detection and response than most PeMS-sourced datasets.

| Dataset | Sensor | Rate | Nodes | Samples | Lane Level | Anomalies |
|---|---|---|---|---|---|---|
| DoTA [31] | Dashcam Video | 10 fps | N/A | 731K | ✗ | ✓ |
| METR-LA [14] | Loop Detector | 5 min | 207 | 7.09M | ✗ | ✗ |
| PEMS-BAY [14] | Loop Detector | 5 min | 325 | 16.94M | ✗ | ✗ |
| ATTAIN LA [19] | Loop Detector | 5 min | 223 | 822K | ✗ | ✓ |
| **FT-AED (Ours)** | **Radar** | **30 sec.** | **196** | **3.76M** | ✓ | ✓ |

However, none of these datasets contain any anomaly information. To perform anomaly detection, researchers need to go through the process of manually finding and matching external report logs to sensor data [16, 4, 19] or even generate anomalies themselves [15]. Some papers have done this work and released their processed subset of the PeMS data source, such as the ATTAIN LA data [19]. Even after doing so, data is typically aggregated every 5 minutes, which may not be granular enough for real-world applications requiring fast anomaly responses.

To address the limitations of current datasets, we collect and release the Freeway Traffic Anomalous Event Detection (FT-AED) dataset. The FT-AED dataset is designed for anomaly detection, focusing on a month of weekday morning rush hour traffic where anomalies, such as crashes, are more likely. Data is captured at the lane level every 30 seconds, enabling quick anomaly detection. See Table 1 for an overview of our dataset in comparison to other freeway anomaly detection datasets. Note that the DoTA dataset is not the only vision-based traffic anomaly detection dataset, but it is selected as a representative dataset from that category, as the focus of this paper is not on video anomaly detection.

## 2.2 Problem Formulation

In this paper, we propose a dataset and form an initial benchmark of baseline methods for the problem of **lane-level freeway anomaly detection**. As we have sensors at each lane and milemarker that collect data over time, we formulate this as a node-level graph anomaly detection problem.

Consider a graph $G = (V, E)$ with a set of nodes, $V = \{v_1, \ldots, v_N\}$, and a set of directed or undirected edges, $E$, represented using an adjacency matrix. Each node has features corresponding to sensor data at a specific lane and milemarker, $v_i \in \mathbb{R}^d$. For our dataset, $N = 49$ and $d = 3$. Then, the goal of node-level graph anomaly detection is to create a mapping, $f : G_t \to \{0, 1\}_t^N$, that determines whether each node is anomalous at time $t$.

This problem is particularly challenging because the true anomaly labels are not known. Additionally, even with some known anomalies, such as crashes, the exact time the crashes occurred is unknown and likely to be delayed. Therefore, while the primary objective is to correctly detect anomalies, **an additional objective is to detect known anomalies, like crashes, before they are reported**.

To this end, we introduce a new metric to minimize called the **Reduction in Reporting Delay**. Since the exact time and location of an incident officially reported by the traffic management center are unknown, this metric measures how much quicker an anomaly is detected. It is defined as follows:

$$\textbf{Reduction in Reporting Delay} = t_{\text{detected}} - t_{\text{incident}} \tag{1}$$

where $t_{\text{detected}}$ is the time an anomaly was detected and $t_{\text{incident}}$ is the time an incident was officially reported. In practice, determining the time an anomaly was detected requires assumptions about the maximum delay in reporting and how long an incident can last. Additionally, since incidents can impact traffic behavior at locations other than where the incident occurs, this metric does not consider the specific node where the anomaly was detected.

## 3 FT-AED Dataset

In this paper, we present the Freeway Traffic Anomalous Event Detection (FT-AED) dataset consisting of high-fidelity traffic measurements derived from Radar Detection Systems (RDS). Our dataset

is one of the first large-scale freeway traffic anomaly detection datasets and offers the potential for real-time incident response and active traffic management. Our dataset consists of traffic data recorded by 49 sensors placed along the Interstate 24 (I-24) corridor, stretching from Murfreesboro to downtown Nashville. This dataset encapsulates traffic states for every workday in October, captured at 30-second intervals. Additionally, we utilize a complementary dataset of event reports, sourced from the Tennessee Department of Transportation Traffic Management Center. These reports are compiled by officers dedicated to monitoring road conditions and identifying unusual occurrences. This dataset not only serves as a benchmark but also paves the way for future research in traffic management and incident response.

## 3.1 Data Collection

To create our anomaly detection dataset, we gathered sensor data from a real-world freeway and incident reports from the traffic management center managing that same freeway.

### 3.1.1 Radar Detection Systems

To detect anomalous incidents on the freeway, we need to represent normal and abnormal traffic behavior using data. In this project, we deployed Radar Detection Systems (RDS), which have been shown to measure traffic speed accurately [9], approximately 0.3 miles apart along an 18-mile stretch of Interstate 24 heading toward Nashville, Tennessee. An image showing the locations of these sensors can be seen in Figure 1. These 49 sensors collected traffic speed, occupancy, and volume data for the four interstate lanes every 30 seconds. We limited the data collected to the peak morning traffic hours of 4:00 am to 12:00 pm during workdays in October 2023. This time window was chosen to limit the focus of anomaly detection to when crashes are most likely to occur. With 8 hours of data from each day every 30 seconds, we have over 3.7 million data points across the 196 nodes.

At each lane and milemarker, we collected sensor data and metadata useful for analysis. The features collected from the RDS sensors are shown below.

- **time_unix**: the time the measurement was taken.
- **milemarker**: location of sensor on the interstate. This comes from the Tennessee Department of Transportation's coordination system and is consistent with what is labeled on the road
- **lane**: the lane number. There are four lanes, where lane 1 is the left-most lane.
- **speed**: 30-second average speed of the vehicles passing through the area.
- **volume**: 30-second average number of vehicles passing through the area.
- **occupancy**: 30-second average percentage of time that the detection zone of the radar sensor is occupied by a vehicle.

### 3.1.2 Incident Labels

With millions of data points across sensor features, it is difficult to define and label all possible anomalies. However, some anomalous incidents of high importance, such as crashes, are tracked by the Tennessee Department of Transportation Traffic Management Center. We sourced the incident logs corresponding to October 2023. These logs contained raw, largely unstructured text detailing each reported crash for the month. We manually parsed these logs, recording the time of each crash. This process uncovered 42 crashes.

Despite their utility, labels from these incident logs have limitations: (1) owing to their reliance on manual reporting, there is an inherent delay in the documentation of events, with the extent of this delay being uncertain, (2) the data structure is somewhat unorganized, being primarily chronological in its event handling; as a result, the recorded end times of incidents may not accurately reflect the actual resolution of the events, complicating their usage, (3) the dataset is susceptible to errors due to the human element in data entry. Despite these drawbacks, the strength of the dataset lies in its ability to offer extensive information and insights upon retrospective analysis of detected anomalies, thereby contributing significantly to the understanding of these events.

Along with labels from event logs, we annotated additional possible anomaly labels using expert knowledge. These labels do not necessarily correspond with crashes and do not suffer from the delayed reporting that the event logs do. We employed space-time diagrams, a frequently used tool for visualizing traffic speed patterns. Through these diagrams, we were able to identify and label

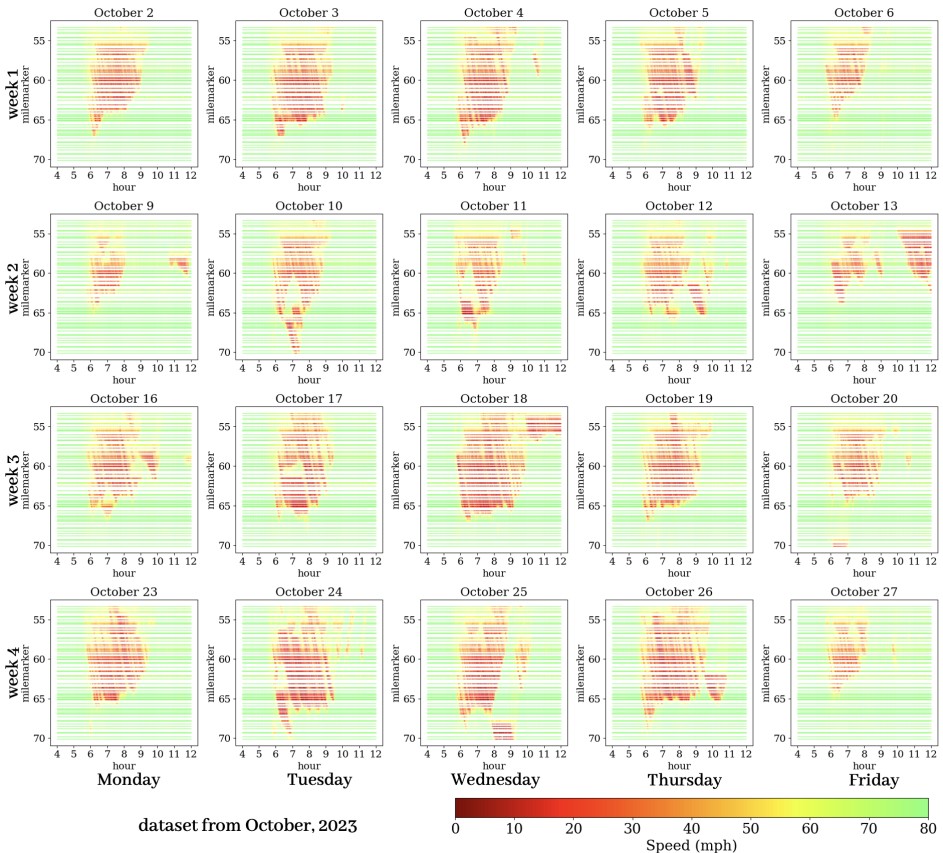

Figure 2: *Freeway traffic dynamics*: October, 2023 weekday lane 1 (high-occupancy vehicle (HOV) lane, often refers to the leftmost lane) speed data visualization for the Westbound (the direction to downtown Nashville) of I-24 section from road reference marker mile 71 to 53, morning peak hours from 4AM and 12PM, sensors are deployed with about a 0.3 to 0.4 mile interval on freeway.

two types of inspected anomalous events not reported in the event records. See Section A.4 in the Appendix for more information. This process led to 19 more labeled anomalies. In our analysis we focus on crash detection, using only the crashes that were officially reported for validation. We use these additional labels to ensure our training data is free of anomalies, a common step in autoencoder-based anomaly detection. By integrating both official reporting logs and diagram analysis, we ensured a comprehensive and accurate labeling of traffic anomalies in the RDS data. An ablation showing the impact of missed anomaly labels is provided in the Appendix (Section A.9).

### 3.2 Data Processing

Over the month of data collection, some sensor data was missing. To ensure our data was compatible with anomaly detection methods that cannot handle missing data, we imputed missing values. To impute speed values we used a domain-specific imputation method that tries to enforce the wave-like patterns in traffic speeds. See Section A.2 in the Appendix for more information. The cleaned speed values for the whole month are shown in the time-space diagrams in Figure 2. For the volume and occupancy features, we used a simple local averaging.

As the goal of the dataset is to perform node-level anomaly detection, graph-based methods can naturally be applied. We propose a baseline graph structure to facilitate easy dataset use for future research. We represent each lane at each milemarker as a node. We connect nodes to adjacent nodes at the same milemarker with the intuition that cars can change lanes at the same milemarker. We connect nodes to all lanes at milemarkers ahead and behind since cars could have changes to any of the four lanes in the 0.3 mile gap between sensors. We formulate this as an undirected unweighted

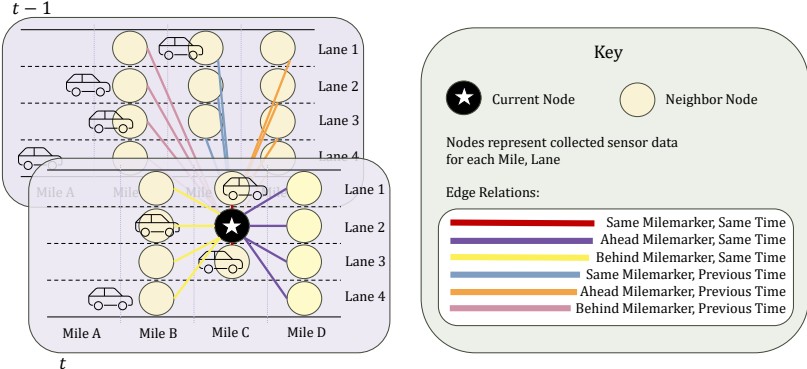

Figure 3: Relational spatiotemporal graph design, illustrated over an example freeway. The full graph is made by forming these connections for all nodes across the time horizon.

graph, making no assumptions about which direction the impact of anomalies flows. A visualization of this graph during a high traffic time can be seen in Figure 8 in the Appendix.

## 3.3 Dataset Usage

The full month of cleaned data is publicly released at the dataset link on our project page `https://vu.edu/ft-aed`. In this repository and the supplementary materials of this paper, there is a dataset card describing additional information for usage. Additionally, we provide a `Demo.ipynb` notebook showing how to import the data and put it into a graph structure for machine learning tasks.

## 4 Experiments

To showcase the potential for our dataset to be used to design and evaluate algorithms for freeway traffic anomaly detection, we conducted initial experiments on our dataset, shown in this section. Numerous additional experiments and more details are also presented in the Appendix.

### 4.1 Additional Preprocessing

Although the general preprocessing steps were described in Section 3, we performed additional preprocessing for our experiments.

**Removing Anomalies.** A common and powerful approach to anomaly detection classifies anomalies based on the reconstruction error of an autoencoder. An autoencoder is trained to reconstruct nominal input. At test time, a point is classified as an anomaly if it is reconstructed poorly since the autoencoder learned to reconstruct nominal points well. We applied this principle to detect incidents and reduce the crash reporting delay. To ensure the training dataset was free of anomalies, we made the following conservative assumptions.

- If a crash has been reported, there could be up to a 30-minute delay in reporting. The impacts of the crash could be present in the data for up to two hours after the crash is reported.
- If an anomaly has been manually labeled, there is no delay. The impacts of the anomaly could be present in the data for up to two hours after the anomaly has been labeled.

**Temporal and Relational Spatiotemporal Graph Design**. Although the primary graph design presented in Section 3.2 describes the network for a single time snapshot, the temporal evolution of the network may also be important. To that end, we designed two additional modifications to the proposed graph structure. First, we designed a **graph time series**. To do this, along with the current graph, we collected the $k$ previous graphs into a time window $= \{G_t, G_{t-1}, \ldots, G_{t-k}\}$. These time windows allow a spatiotemporal model to learn spatial and temporal relationships in the data. Second, we designed a **relational spatiotemporal graph**. First, we connected all $k$ graphs into a single graph, allowing time to be considered by spatial methods. Additionally, each edge was given a relational class describing how the two connected nodes are related. This allows a graph algorithm to learn

different weights for each relation type, passing information differently on the edge category. This relational spatiotemporal graph is illustrated from the perspective of a single node in Figure 3.

## 4.2 Model Training

With the data processed, we trained anomaly detection models on the data. The models we trained are described below.

- **STG-RGCN AE** - Relational Graph Convolutional Network Autoencoder. This model uses the relational spatiotemporal graph of the freeway network as input and Relational Graph Convolution [25] blocks.
- **STG-GAT AE** - Graph Attention Network Autoencoder. This model uses the spatiotemporal graph without edge relations as input and Graph Attention [29] blocks for spatiotemporal learning.
- **GCN-LSTM AE** - Graph Convolutional Network Autoencoder with temporal aggregation in the latent space using a Long Short-Term Memory Network. It accepts a time series of spatial graphs as input. It uses Graph Convolution [10] blocks for spatial processing and LSTMs for temporal processing.
- **GCN AE** - Graph Convolutional Network Autoencoder. This is a special case of the STG-RGCN AE without edge relations and using only the current graph as input.
- **Transformer AE** - Transformer Autoencoder. It processes the time series of node features, treating each node as independent. It uses temporal but not spatial features.
- **MLP AE** - Multi-layer Perceptron Autoencoder. This is a standard autoencoder that treats each node as independent and does not consider temporal or spatial features. It reconstructs each node using its own features.

To train the models, we minimized the reconstruction error of the node features of the current graph. The loss function is shown below, where $X_t$ are all the node features captured from the freeway at time $t$ and $\hat{X}_t$ are the reconstructions of those features given by the model.

$$\mathcal{L}_{\text{mse}} = \underset{X}{\mathbb{E}} \, ||X_t - \hat{X}_t||_2^2, \tag{2}$$

We also optimized the hyperparameters of each of the trained models. Using the optimized hyperparameters, we trained each model on 14 days of morning data. We kept 5 days for validation and left 1 day out due to excessive, uncertain anomalies. With a trained model, anomalies were detected for each node in the network using the following:

$$\text{anomalies}(X_t, \hat{X}_t) = (X_t - \hat{X}_t)^2 > T, \tag{3}$$

where $T$ is a vector of thresholds, one for each node. We set this as the maximum squared reconstruction error on the training set for each node. Therefore, no samples in the training data were classified as anomalies. However, we tuned this threshold to control the false positive rate in our experiments. Additional details regarding hyperparameter optimization, figures showing model architectures, and formal definitions of the graph operations are shown in the Appendix Section A.6.

## 4.3 Anomaly Detection Results

To evaluate the anomaly detection performance, we computed the following metrics after applying the trained models to the 5 days of validation data.

- **Reduction in Reporting Delay**: see Equation 1. We consider this the most important metric. A lower reduction in reporting delay implies that the model is detecting anomalies sooner than they are noticed and recorded by the traffic management center. We calculate this within a 15-minute window before and after a crash occurs. Lower is better.
- **Miss Percentage**: the percentage of crashes that were not detected as anomalies within 15 minutes of being officially reported. Lower is better.
- **Reconstruction Error**: see Equation 2. The mean squared reconstruction error on the data free of anomalies. A model that has better learned to reconstruct the node features may have better learned general nominal traffic behavior and may more accurately detect anomalies, especially in cases with heavy traffic congestion. Lower is better.

Table 2: Overall relative performance of each proposed method. Note that Reporting Delay (mean $\pm$ standard deviation) was computed using an anomaly threshold that fixed the FPR at $5\%$. Reconstruction Error on anomaly-free validation data.

| Autoencoder Model | Reporting Delay | Miss Percentage | Recons. MSE | AUC |
|---|---|---|---|---|
| STG-RGCN | $-8.95 \pm 7.59$ | **16.67**% | 0.0102 | 0.67 |
| STG-GAT | $-7.75 \pm 6.58$ | **16.67**% | **0.0089** | 0.68 |
| GCN-LSTM | $-6.83 \pm 8.18$ | 25% | 0.0119 | 0.65 |
| GCN | $\mathbf{-10.20 \pm 5.98}$ | 25% | 0.0095 | **0.70** |
| Transformer | $+2.42 \pm 8.35$ | 41.67% | 0.0312 | 0.60 |
| MLP | $-6.86 \pm 6.87$ | 41.67% | 0.0122 | 0.62 |

- **False Positive Rate (FPR)**: the percentage of detected anomalies that were not anomalies. To determine whether a predicted anomalous time was a True Positive or True Negative, we assume that there was no longer than a 15-minute delay in reporting and that the impacts of a crash can last up to 2 hours, similar to [19, 4]. These assumptions are purposefully conservative and impact the usefulness of typical binary classification metrics. See Section 5 for further discussion. Lower is better.
- **Area Under the Curve (AUC)**: the area under the receiver operating characteristic curve, demonstrating the tradeoff between True Positive and False Positive Rate at all anomaly thresholds. Higher is better.

The overall relative performance of each method can be seen in Table 2. The **GCN Autoencoder with no temporal information (GCN) achieved the lowest average detection delay of the crashes detected** with the lowest variance. It detected crashes well before they were officially reported, over 10 minutes on average. It also achieved a higher AUC than the other methods. This implies that the temporal relationships between nodes was less important than the spatial relationships. Considering the task of an Autoencoder, this makes sense. Most of the information needed to reconstruct the current graph was present in the current graph. More evidence for this point is shown by the poor performance of the Transformer Autoencoder. A simpler MLP vanilla Autoencoder detected crashes faster on average than the more complex Transformer model. (Though neither of their reconstructions were reasonable due to having to project from a 3-dimensional latent space to a 2-dimensional latent space.) Additionally, the GCN model missed a larger percentage of the crashes than two of the other methods, implying that temporal information may be necessary to detect some crashes.

Next, we can consider the Reconstruction MSE on the nominal data from the 5-day validation set. The GAT Autoencoder with the spatiotemporal graph (STG-GAT) achieved the lowest reconstruction error. Since this was the training task, this model appears to have learned and generalized best, but the practical differences among the MSEs are not clear from this metric alone. The RGCN Autoencoder with the spatiotemporal graph (STG-RGCN) had the second-lowest detection delay but had a higher variance in its detection delay. Finally, the GCN Autoencoder with an LSTM for temporal aggregation (GCN-LSTM) had the highest detection delay with the highest variance, highest Reconstruction Error, and lowest AUC. In this case, it was the worst model in all metrics. **Moving the temporal aspect of the framework into the data instead of the model improved the performance.** At the same time, it improved the parallelization of the computations by removing the recurrent layers.

### 4.4 Case Study

Next, we can analyze a case study of a specific crash, chosen from the validation set. On October 11, 2023, the incident logs from the traffic management system reported, *"Crash in RUTHERFORD county going Westbound on Interstate 24 beyond MILE MARKER 63.4 Last updated 10/11/2023 6:24:31 AM"* and *"Crash in RUTHERFORD county going Westbound on Interstate 24 beyond MILE MARKER 63.4 with **Left lane blocked**"* was reported just over a minute later.

We ran our STG-GAT AE model with an anomaly threshold that fixed the validation FPR at $10\%$ on October 11, 2023 morning data. In Figure 4a, the vehicle speeds over time and space in the left lane are shown. Our model detected a large number of anomalies beyond milemarker 63.5 starting at 6:07:30 AM and ending at 6:42:00 AM. **We consistently detected the crash at the correct location 17 minutes before it was officially reported.** Additionally, qualitatively, there were a small number of false positives in the plot. In fact, two of the minutes of data were incorrectly determined to be false

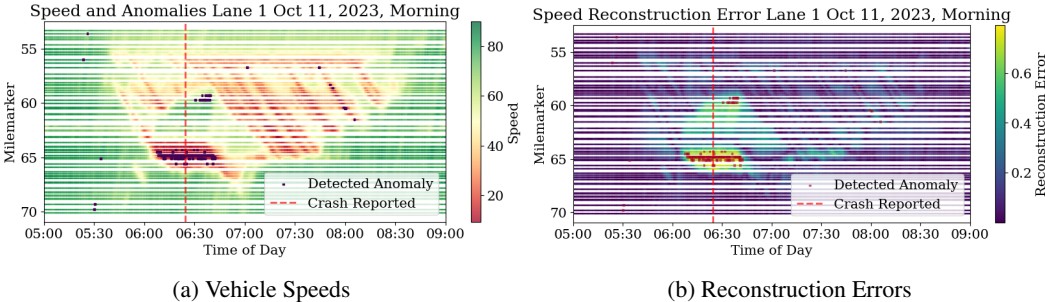

(a) Vehicle Speeds          (b) Reconstruction Errors

Figure 4: Case study. Crash detection and reporting for the morning of October 11, 2023, using STG-GAT AE. The threshold was chosen such that there was a $10\%$ validation FPR.

positives when they were directly connected to the crash because they were more than 15 minutes before it was reported. The reconstruction errors are shown in Figure 4b. From this Figure, we can see that the points around the crash were reconstructed worse than the nominal points. This is further evidence of the appropriateness of reconstruction-based approaches for this problem.

## 5 Future Directions and Limitations

From our dataset design and baseline benchmark, we have opened the door to future research. In this section, we briefly detail future research directions to build on the dataset challenges and limitations.

**Anomalies are detected at the lane level, but validation is not done at the lane level.** As shown in Figure 4, our proposed methods detected anomalies accurately at the lane level. However, since most of the labeled anomalies our dataset had were crashes, the impacts of these may cause anomalous traffic behavior in locations other than the crash site (e.g., everywhere behind the crash is now stuck in traffic). Future methods can research how to validate the location of detected anomalies.

**There is a need for an overall performance metric besides AUC.** Since anomalies were recorded as instantaneous points in time and the definition of a true positive or true negative are unknown for complex anomalies like crashes, AUC can be a misleading metric. Future research can evaluate different metrics to determine which metric corresponds best with practical performance.

**The speed reconstructions lose some of the dynamics of the traffic system.** Comparing the reconstructions in Figure 4 to Figure 2, the reconstructions are reasonable. However, upon further inspection, they appear to lose some of the "wave" patterns present in the real data. This may cause subtle anomalies, especially when crashes occur, to go undetected. Therefore, future methods can explore how to reconstruct the speed more accurately without losing the autoencoder bottleneck. We suspect enforcing the wave-like behavior as a regularization term in the loss function would help.

**More anomaly detection approaches should be benchmarked.** In our baseline experiments, we only used reconstruction-based autoencoder anomaly detection approaches. This was due to the uncertainty in the data labels; supervised approaches may fail. However, other rule-based and unsupervised anomaly detection methods exist. See Appendix A.8 for an initial exploration into the performance of clustering and rule-based methods. Future research should use our dataset as a benchmark for more anomaly detection algorithms.

## 6 Conclusion

In this paper, we presented the Freeway Traffic Anomalous Event Detection (FT-AED) dataset, the first large-scale real-world dataset focused on lane-level freeway anomaly detection. The FT-AED dataset comprises over 3.7 million sensor measurements, capturing the vehicle speed, occupancy, and volume in 4 lanes along 49 milemarkers over the weekday mornings of October 2023. We sourced crash records from the Tennessee Department of Transportation Traffic Management Center as ground truth anomaly labels. Due to human delays and errors, there is uncertainty in the reported crash time, leading to a unique challenge in training and validating anomaly detection models. We also manually labeled additional potential anomalies based on expert inspection of the vehicle speed

profiles. To assess the ability of our dataset to be used as a benchmark, we trained and validated 6 autoencoder-based anomaly detection methods. In this, we found that GNN Autoencoders detected anomalies better than purely temporal or feature-based models and that introducing a spatiotemporal graph could help reduce the number of missed crashes. We detected crashes minutes before they were officially reported, highlighting a specific case where our methods detected a crash at the correct location 17 minutes before it was reported. We hope that developing and releasing this dataset will lead to further advancements in anomaly detection methods and automate incident response so delays like this can be avoided.

## Acknowledgment

This work is supported by a grant from the U.S. Department of Transportation Grant Number 693JJ22140000Z44ATNREG3202 and the Tennessee Department of Transportation under the grant RES2023-20. This material is based upon work supported by the National Science Foundation under Grant No. CNS-2135579 (Work). The U.S. Government assumes no liability for the contents or use thereof. We would also like to acknowledge the Tennessee Department of Transportation (TDOT) for providing the data used in this research.

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

# A  Appendix

## A.1  Crash Example

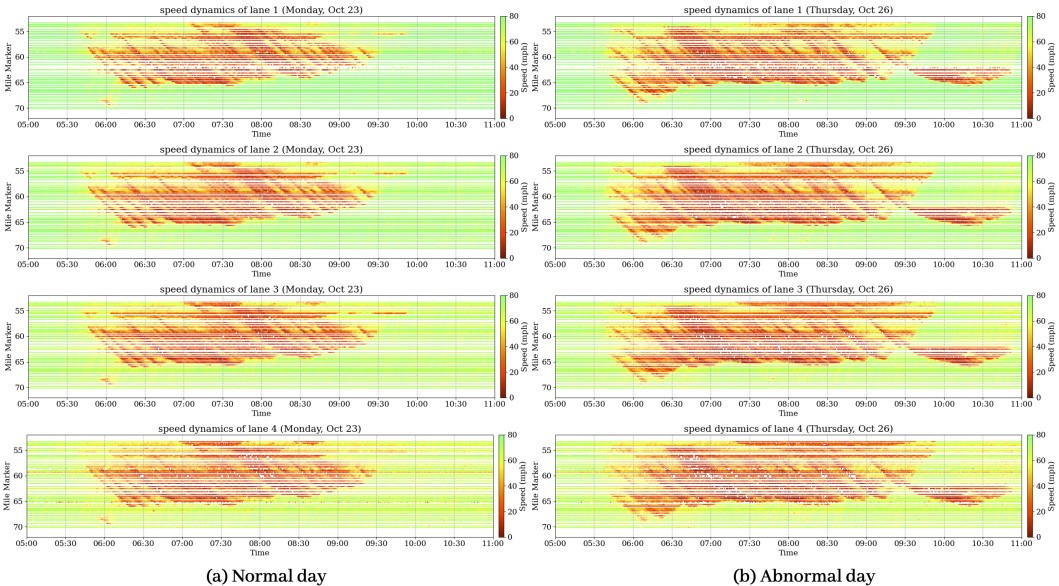

(a) Normal day           (b) Abnormal day

Figure 5: **Lane-level data comparison**: the left is Monday, a normal workday, the right is Thursday, a very clear crash can be seen by the triangle shape present around 9:30 AM in the right figures.

An example of how a crash impacts traffic behavior can be seen in Figure 5. In the left column, the vehicle speeds on a normal day are plotted. The X axis represents the time of day. The Y axis represents the milemarker (location) of the sensor. Each row of the figure grid is a different lane. The colors of the points represent the speeds. On a normal day, traffic contests on the freeway heading toward the city at around 5:45 AM. This congestion lasts until around 9:00 AM.

On an abnormal day, shown in the right column, there is an additional triangular-shaped structure starting at 9:30 AM. This triangle indicates a crash. Around milemarker 62, the vehicle speeds have drastically decreased. Over time, traffic at higher milemarkers (behind the vehicle that has crashed) also slows down. When the crash clears, traffic speeds back up. Many crashes can be manually detected **after they happen** from observations like these.

## A.2  Imputation

Due to unexpected issues in the sensor network, imputation is needed for missing data. In this paper, the *Adaptive Smoothing Method* (ASM) is used for imputation. ASM, developed by [28], is a widely used algorithm for smoothing and imputing data to create a continuous spatiotemporal mean speed field. The core concept of ASM is the division of the mean speed field into two components: a free-flow field and a congested field. This separation allows the method to leverage the distinct and consistent propagation velocities characteristic of free-flow and congested traffic. Specifically, ASM smooths and imputes data in free-flow conditions along lines that match the free-flow speed of traffic, while in congestion, it smooths and imputes data along lines corresponding to the backward propagating wave speed. For a detailed mathematical explanation of the imputation method, refer to [28].

## A.3  Occupancy and Volume

A similar figure to Figure 2 for occupancy and volume are shown in Figures 6 and 7, respectively. All three features are passed to the trained models.

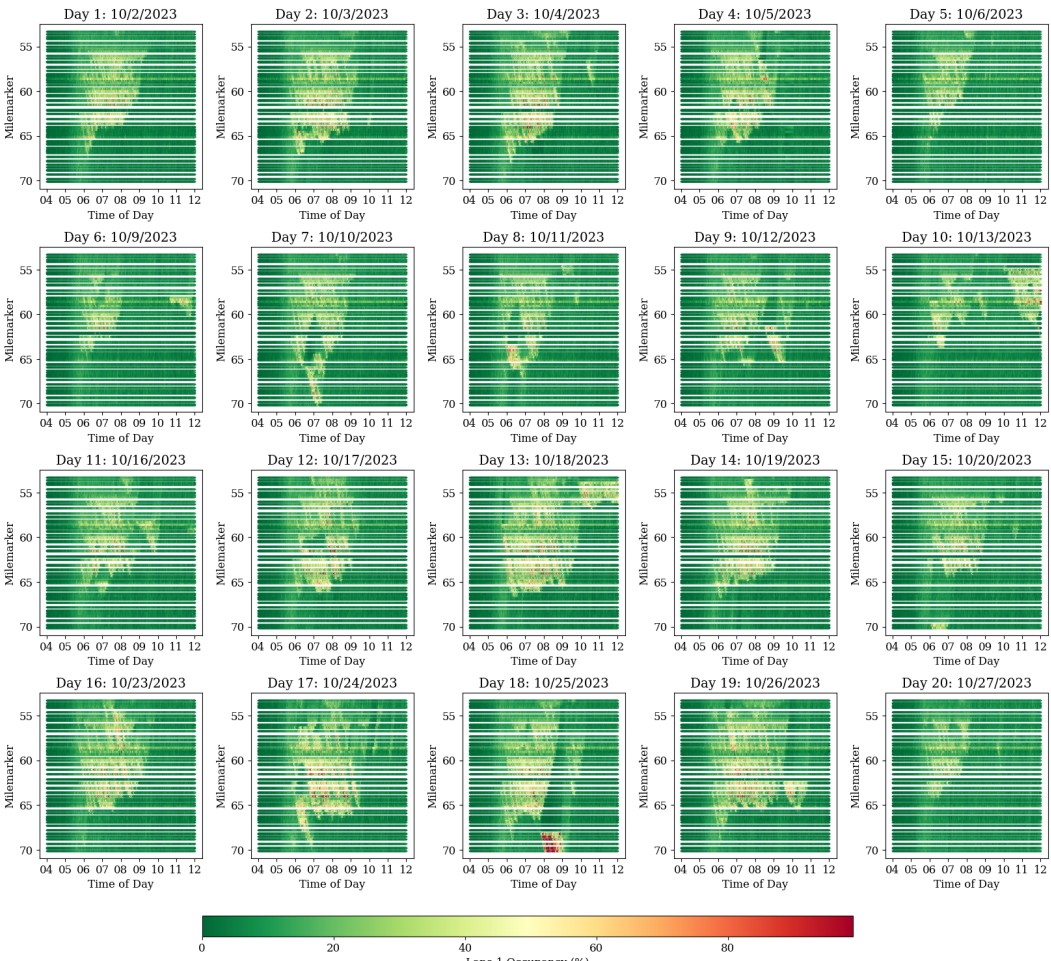

Figure 6: *Freeway traffic dynamics dynamics*: October, 2023 weekday lane 1 (high-occupancy vehicle (HOV) lane, often refers to the leftmost lane) **occupancy** data visualization. Occupancy is the average percentage of time that the detection zone of the sensor is occupied by a vehicle.

## A.4 Manual Incident Labeling

To manually label additional anomalies, we visually inspected the speed time-space diagrams for the following

- **triangular traffic patterns**: these patterns emerge due to sudden changes in traffic flow following an incident. They appear as distinct triangular shapes in the space-time diagrams and are crucial for identifying under-reported events.

- **moving bottlenecks**: slow-moving vehicles that obstruct traffic lanes. In the diagrams, these bottlenecks create patterns that move slowly along with the direction of traffic flow (see Figure 2, October 24, 10 AM milemarker 60 for details).

## A.5 Graph Structure

An illustration of our proposed basic graph structure described in Section 3 for a single time instance can be seen in Figure 8. At 7 AM, the vehicles have slowed down, indicated by the color of the nodes. This graph structure can serve as a baseline when using our dataset. We extend our graph structure to a spatiotemporal relational graph in Figure 3.

Lane 1 Volume

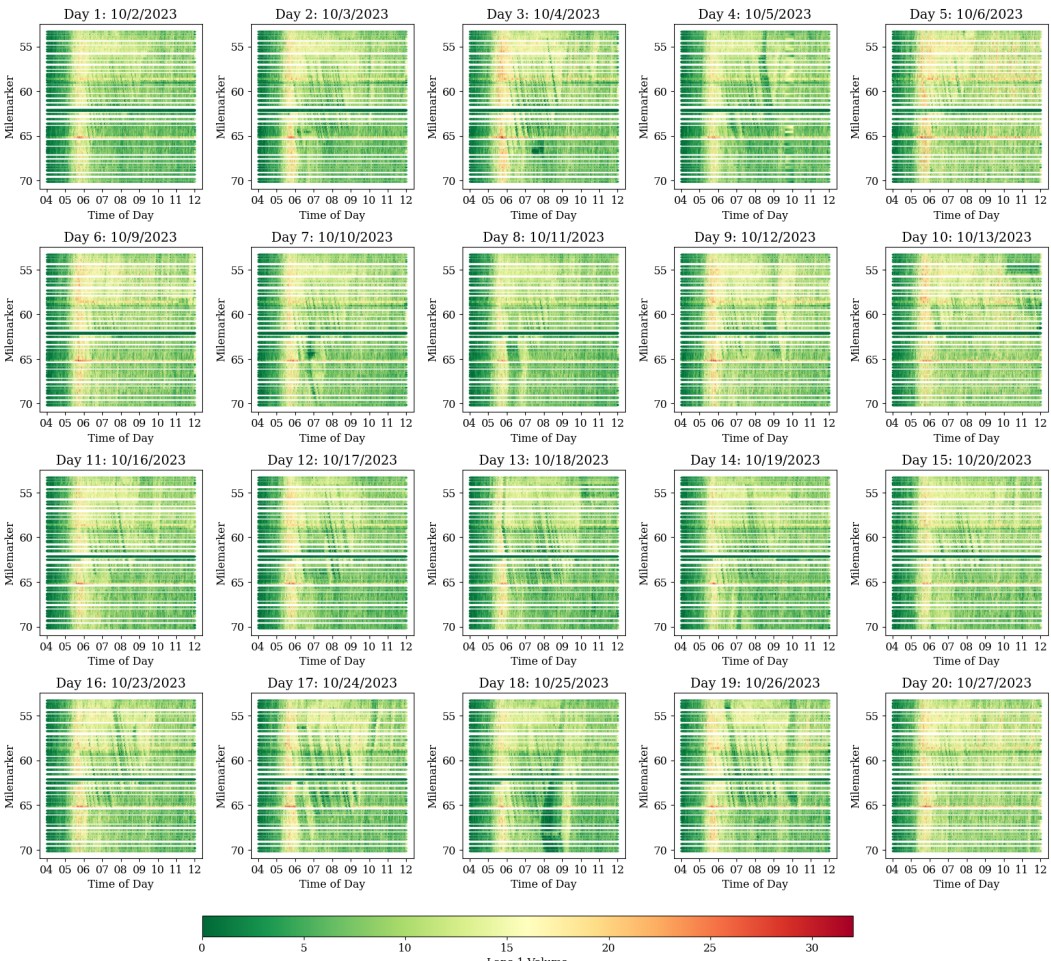

Figure 7: *Freeway traffic dynamics dynamics*: October, 2023 weekday lane 1 (high-occupancy vehicle (HOV) lane, often refers to the leftmost lane) **volume** data visualization. Volume is the average number of vehicles passing through the area.

## A.6 Model Details

The models we evaluate in this paper are autoencoders. An autoencoder consists of an encoder $\phi$ and decoder $\psi$. The encoder maps the input observations to a lower-dimensional latent representation $z \in \mathbb{R}^k$, $\phi : \mathbb{R}^n \to \mathbb{R}^k$, where $k < n$. The decoder maps the reduced dimensional $z$ to the original input space $\psi : \mathbb{R}^k \to \mathbb{R}^n$. The autoencoder computation can be represented as $\psi(\phi(x))$, where $x \in \mathbb{R}^n$ is some input data point. Autoencoders are trained by minimizing the difference between the original input point and the reconstructed input point, typically quantified using MSE (Equation 2). Autoencoders are applied to anomaly detection by exploiting the fact that deep learning models often fail to generalize to unseen tasks. By training an autoencoder to reconstruct data free of anomalies, we expect it to reconstruct unseen anomalous data worse than seen nominal data. Since we do not have trustworthy labels for crashes in our dataset, this unsupervised formulation can be powerful. An overview of our proposed autoencoder anomaly detection framework is shown in Figure 9.

In our experiments, we alter the input data format and encoder and decoder architectures. The simplest architecture we use is a vanilla multi-layer perception autoencoder (**MLP AE**). In this case, each location is treated as independent, ignoring possible spatial or temporal dependencies between nodes. We use a two-layer MLP to project the 3-dimensional input into a 2-dimensional latent space.

# Traffic Speed (7 am)

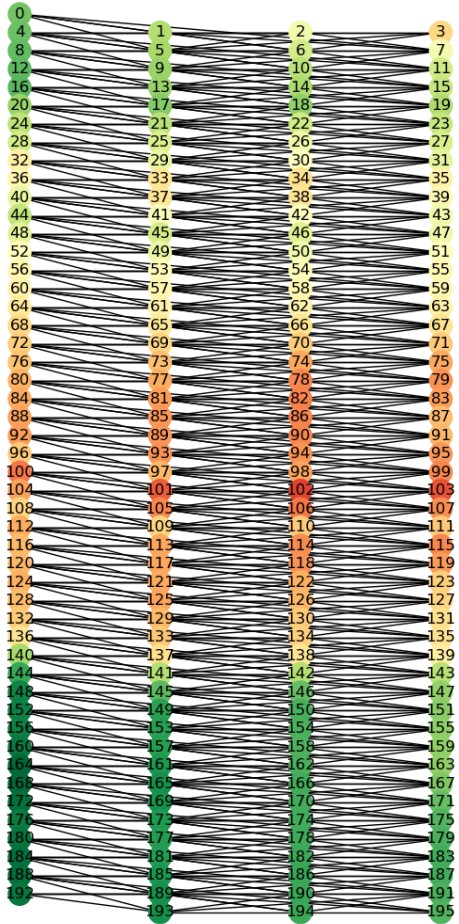

Figure 8: Graph structure example. 7 am, Monday, October 23 on I-24 Westbound. Rows are different milemarkers, columns are different lanes. Red indicates lower average speed at that node.

Then, the original features are reconstructed. To enforce a bottleneck in the latent space, the latent vector can only be 1 or 2-dimensional, heavily limiting the expressiveness in the latent space.

Next, we develop an autoencoder that considers temporal dependencies in a node. We use a Transformer autoencoder (**Transformer AE**) for this purpose. The Transformer AE treats each node as an independent time series, ignoring spatial dependencies. A time window of recent points is passed to the Transformer AE. It projects this to a 2-dimensional latent vector. This is then projected back to a 3-dimensional reconstruction of the speed occupancy and volume at the current time.

### A.6.1   Graph-based Autoencoders

To introduce spatial relationships, we use graph neural network layers. The simplest form of these that we use is the graph convolutional (GCN) operator. A GCN is a message-passing operator that has the following form. (Notation from the original paper [10].)

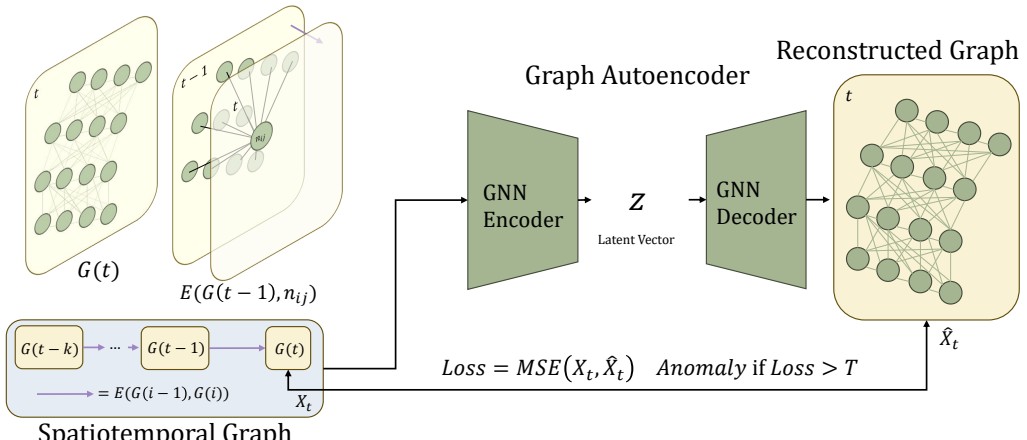

Figure 9: Overview of proposed incident detection approach.

$$H^{(l+1)} = \sigma(\tilde{D}^{-\frac{1}{2}}\tilde{A}\tilde{D}^{-\frac{1}{2}}H^{(l)}W^{(l)}) \tag{4}$$

In this equation, $\tilde{A} = A + I_N$ is the self-connected adjacency matrix, $\tilde{D}_{ii} = \sum_j \tilde{A}_{ij}$, $H^{(l)}$ are activations in layer $l$, $W^{(l)}$ are weights in layer $l$, and $\sigma$ is a nonlinear activation function.

The **GCN** autoencoder uses this operator. In this model, we use a GCN encoder and decoder. We accept the current network graph displayed in Figure 8 as input. The encoder uses the operation from Equation 4 along with global mean pooling to project the graph to a latent vector of dimension $k$. The latent vector is linearly projected into a larger vector of size $k\times$ num_nodes. We then recreate the graph, sequentially allocating each of the nodes a $k$ dimensional latent vector. Then, the GCN decoder reconstructs the input graph.

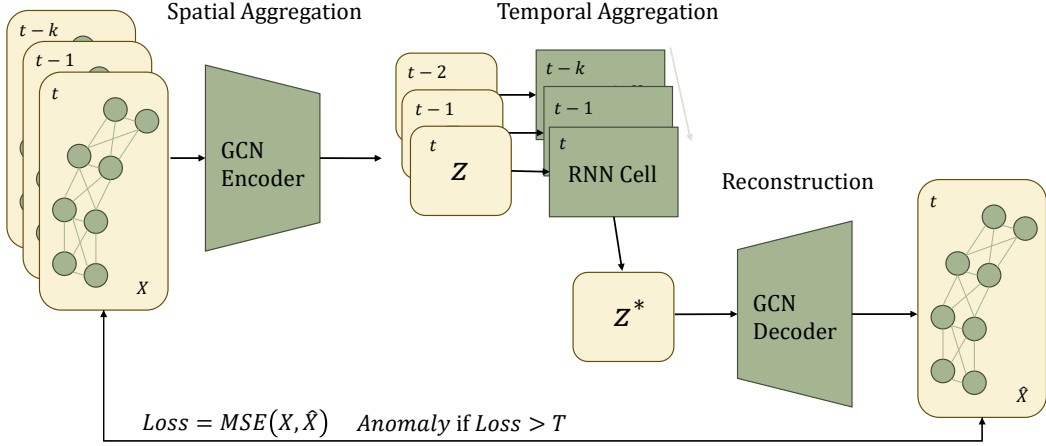

Figure 10: GCN-LSTM AE.

The GCN autoencoder uses spatial information to detect anomalies. To introduce temporal informa-tion, we created variations on spatiotemporal autoencoders. The first variation, the **GCN-LSTM** accepts a time window of graphs as input. Each graph is encoded independently using a GCN encoder, creating a window of latent vectors. Then, this latent vector time series is fed to a long short-term (LSTM) recurrent neural network model to process temporal relationships. Then, this latent vector is decoded in the same way as in the GCN autoencoder to reconstruct the current network graph. See Figure 10 for an overview of this model.

Instead of encoding temporal dependencies in the latent space of the model, we also inject it directly into the data by building a spatiotemporal graph as described in Section 4.1 and Figure 3. With this graph structure, we train another GCN (**STG-GCN**) autoencoder.

Aside from GCNs, we use two other graph neural network models for encoding and decoding the spatiotemporal graph. First, we use a graph attention network (GAT). Graph attention networks add self-attention to graph neural network message passing. A GAT is defined as follows [29].

First, the attention coefficients between nodes are computed.

$$e_{ij} = a(Wh_i, Wh_j) \tag{5}$$

where $W$ is a weight matrix applied to every node, $h_i$ is is the features of node $i$, and $a$ is an attention mechanism. Then, the coefficients are normalized.

$$\alpha_{ij} = \text{softmax}_j(e_{ij}) = \frac{\exp(e_{ij})}{\sum_{k \in \mathcal{N}_i} \exp(e_{ik})} \tag{6}$$

where $\mathcal{N}_i$ is the neighborhood of node $i$. Finally, the attention is aggregated and a nonlinearity is applied.

$$h'_i = \sigma \left( \sum_{j \in \mathcal{N}_i} \alpha_{ij} W h_j \right) \tag{7}$$

We employ this GAT operation with our spatiotemporal graph structure in the encoder and decoder of our STG-GAT model. Finally, we also consider that a node might have different kinds of relationships among its neighbors in the spatiotemporal graph. For example, a node from the same time but at an ahead milemarker might be impacted differently by a crash at the current node than one in the past at a milemarker behind. To categorize these relationships, we use relational graph convolutional networks (R-GCN). An R-GCN is defined below [25].

$$h_i^{(l+1)} = \sigma \left( \sum_{r \in \mathcal{R}} \sum_{j \in \mathcal{N}_i^r} \frac{1}{c_{i,r}} W_r^{(l)} h_j^{(l)} + W_0^{(l)} h_i^{(l)} \right) \tag{8}$$

where $r \in \mathcal{R}$ is a relation (a class of edge) and $c_{i,r}$ is a (learned) normalization constant. In our case, the relations are shown in the key of Figure 3. We use this in our **STG-RGCN** model, the same was the GAT was used above, to consider spatial and temporal relationships of different types when learning to reconstruct the current graph.

### A.6.2   Hyperparameters

To tune the hyperparameters of our models, we used the Tree-Structured Parzen Estimator algorithm [3] included in the Python library, Optuna [2]. We chose this algorithm because it performs an intelligent hyperparameter search over both discrete and continuous hyperparameters. We optimized the hyperparameters on a smaller task trained on one morning of crash-free data and validated on a separate morning of crash-free data. The hyperparameters chosen are given below. Note that the hyperparameters of the Transformer were manually tuned due to significantly higher computational demands with worse performance.

Hyperparameters chosen through the hyperparameter optimization can be seen in Table 3. From this, we can observe some interesting patterns. For example, all the models chose to use a small number of GNN layers. Since the graph is small for this task, a simpler model may perform better. This is also evidenced by the hidden dimension never being the maximum of 256. We can also see that both the STG-RGCN and GCN-LSTM models preferred a 4-minute window of historical data (8 30-second timesteps), either in the spatiotemporal graph or in the graph sequence. This emphasizes the idea that the temporal evolution of the road network may be useful in this task. However, the GAT chose a small timestep value of 2. Perhaps the attention operation in the GAT more effectively leverages information from smaller time windows.

Table 3: Optimized hyperparameters.

| Hyperparameter | RGCN | GAT | GCN-LSTM | GCN | Transformer | MLP |
|---|---|---|---|---|---|---|
| Dropout | 0.45 | 0.09 | 0.43 | 0.03 | - | - |
| Hidden Dim | 128 | 128 | 128 | 64 | 64 | 128 |
| Latent Dim | 32 | 256 | 64 | 128 | 2 | 2 |
| Learning Rate | 0.0017 | 0.0004 | 0.0002 | 0.0047 | 0.0001 | 0.0023 |
| Timesteps | 8 | 2 | 8 | - | 10 | - |
| (G)NN Layers | 1 | 1 | 2 | 2 | 2 | 2 |
| LSTM Hidden | - | - | 32 | - | - | - |
| LSTM Layers | - | - | 1 | - | - | - |
| GAT Heads | - | 4 | - | - | - | - |
| Transformer Heads | - | - | - | - | 1 | - |
| Transformer Layers | - | - | - | - | 2 | - |

## A.7 Early Detection Performance

Table 4: FPR impact on Reporting Delay (mean $\pm$ standard deviation) and number of crashes not detected within 15 minutes of being reported (Miss Percentage) for each model.

| Model | Reporting Delay | Miss Percentage |
|---|---|---|
| 1% FPR | | |
| STG-RGCN AE | $-1.56 \pm 8.83$ | 33% |
| STG-GAT AE | $-1.85 \pm 10.01$ | 41.67% |
| GCN-LSTM AE | $-2.29 \pm 8.11$ | 41.67% |
| GCN AE | $-3 \pm 9.56$ | 41.67% |
| 2.5% FPR | | |
| STG-RGCN AE | $-5.81 \pm 8.62$ | 33% |
| STG-GAT AE | $-6.11 \pm 8.81$ | 25% |
| GCN-LSTM AE | $-5.11 \pm 9.94$ | 25% |
| GCN AE | $-6.67 \pm 9.17$ | 25% |
| 5% FPR | | |
| STG-RGCN AE | $-8.95 \pm 7.59$ | 16.67% |
| STG-GAT AE | $-7.75 \pm 6.58$ | 16.67% |
| GCN-LSTM AE | $-6.83 \pm 8.19$ | 25% |
| GCN AE | $-9.78 \pm 6.89$ | 25% |
| 10% FPR | | |
| STG-RGCN AE | $-7.18 \pm 9.62$ | 8.33% |
| STG-GAT AE | $-13.33 \pm 2.56$ | 0% |
| GCN-LSTM AE | $-8.18 \pm 7.73$ | 8.33% |
| GCN AE | $-9.85 \pm 6.50$ | 16.67% |

As previously mentioned, the delay in crash reporting is a major challenge we are trying to address in this paper. By addressing it, future crash labeling can be more accurate and incident response time by the traffic management team could be reduced. In Table 4, we analyzed the impact of changing the anomaly threshold such that we had a fixed percentage of validation false positives on crash reporting. We fixed the threshold such that the false positive rate was in $\{1\%, 2.5\%, 5\%, 10\%\}$. From this Table, we can see that **even with only $1\%$ false positives, when we detected a crash, we detected it before it was reported on average with all methods**. However, with this low FPR, there was a large standard deviation in the Reporting Delay and many crashes were missed. As we allowed larger validation FPRs, the reporting delay and miss percentage decrease. At $5\%$ FPR, the STG-GAT AE had a lower Miss Percentage than the GCN AE, and at $10\%$ FPR, the STG-GAT AE did not miss any crashes and had the lowest reporting delay.

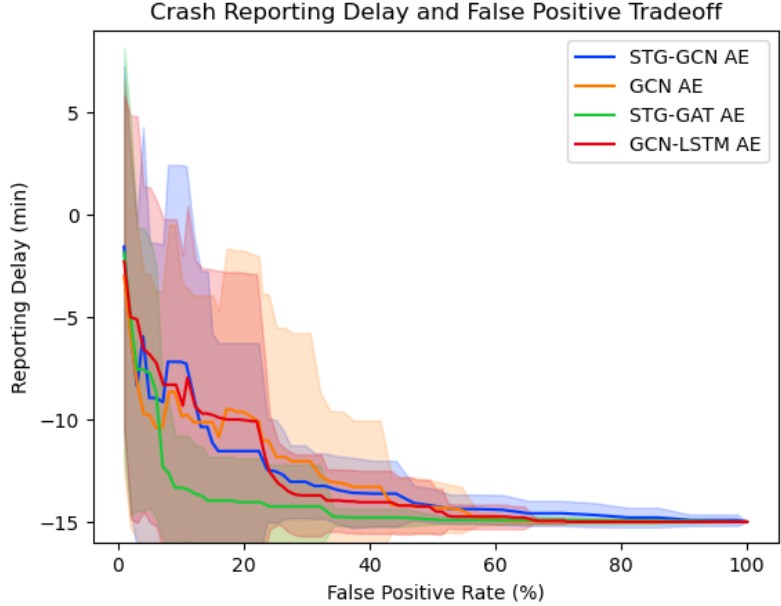

Figure 11: False Positive Rate impact on Crash Detection Delay versus the report log for crashes detected. Jumps are caused by a previously missed crash now being detected.

We can supplement this analysis by looking at Figure 11. As can be seen, for lower FPR, the GCN AE has the lowest average detection delay. As the anomaly threshold decreases and the FPR is allowed to grow, the STG-GAT AE and STG-RGCN AE pass the GCN that does not use temporal information. Additionally, the STG-GAT AE has the lowest detection delay by far of all the models, implying that using the spatiotemporal graph and attention may be better than RGCNs or GCNs in this problem.

Before applying this model in the real world, a user would have to decide a threshold appropriate for their application. For example, an officer at a traffic management center may only have to switch camera views on their computer to check if a crash has happened. In that case, false positives are cheap and they may want to use the STG-GAT AE with the threshold that achieves a $10\%$ FPR to detect crashes much quicker than they would report them. If it is not that easy to verify if crashes have happened, they may choose to use a more conservative threshold that may miss some crashes but still detect crashes before they were reported, on average, such as the GCN AE with $1\%$ FPR. Of course, these thresholds were chosen on our validation data, and may not achieve that FPR in the real world, so more testing would need to be done to see if the FPR and Detection Delay hold on more data.

Finally, we can analyze the distributions of crash detection delays on the full 19-day dataset (still excluding the one day with excessive anomalies), including the crashes during the days of training data. (Note that this dataset could not be used for the above analysis because it would unfairly lower the FPR.) Figure 12 shows the distributions for each method at a threshold that fixes the full FPR at $10\%$. As can be seen, the STG-GAT AE reduces the delay the most on average and at maximum. It misses none of the 44 reported crashes. The ordering then goes STG-RGCN AE, GCN AE, and GCN-LSTM AE. The STG-RGCN AE and GCN AE miss 3/44 crashes and the GCN-LSTM AE misses 4/44 crashes. This further enforces the previously discussed strengths of each model.

### A.8   Additional Anomaly Detection Methods

All the methods explored in this paper were reconstruction-based autoencoders. Future work should develop and evaluate novel anomaly detection approaches to enhance freeway traffic anomalous event detection. However, to get an initial sense of the performance of other methods, we implemented two additional anomaly detection approaches.

The first additional approach we evaluated was K Means clustering. This algorithm is simple, easy to tune, and very interpretable. Future work should try more advanced clustering methods like

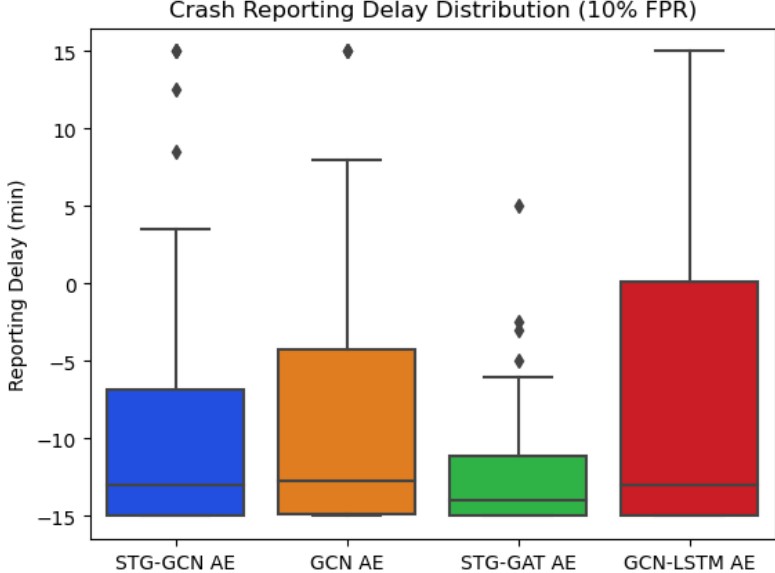

Figure 12: Distributions of crash detection delays on the full dataset under a threshold that achieves a 10% validation false positive rate. Misses are assigned a 15-minute delay to be shown.

density-based or spectral clustering. We increased the number of clusters on the anomaly-free training data until the inertia (within-cluster sum-of-squares) converged. This led to 3 clusters, shown in Figure 13 below. Then, we determined the distance of each test point from the cluster centroids and varied the distance threshold for anomaly detection. As in the main results, we fixed this threshold to demonstrate the performance with a 5% false positive tolerance.

Next, we implemented a widely-used rule-based algorithm, the California algorithm [23]. This simple technique determines if a node is in an anomalous state by observing the upstream and downstream occupancy. It uses predefined thresholds that are known to be hard to tune. It typically "requires the laborious calculation of thresholds for each location where it is installed. In large networks, separate thresholds must be calculated for different road geometries" [20]. As a starting point, we used thresholds from the original paper [23] (T1=8.1, T2=0.313, T3=16.8). Then, we attempted to optimize the three thresholds to fit the 5% false positive tolerance. After only a few iterations, the optimizer was stuck at the original thresholds. These thresholds led to about 8% of the nodes being false positives. However, it detected an anomalous node 99% of the time. Consequentially, it never failed to detect the time a crash was reported. Since ground truth spatial targets for anomalies are not available (and are not desired since incidents can impact the whole road network), this flaw heavily hinders the performance of this algorithm.

The results of both experiments are shown in Table 5 below. The clustering approach leads to reasonable performance, detecting half the crashes before they are reported on average. It can maintain a 5% false positive rate. The California algorithm perfectly detects crashes due to its high false positive rate. We do not consider this to be representative of the best possible performance of the California algorithm. Rather, it highlights the benefits of the other approaches proposed in the paper with thresholds that can easily be tuned. In both cases, more advanced approaches should be benchmarked in future work.

Table 5: Performance of additional anomaly detection methods. Note that the California algorithm almost always detects at least one node as an anomaly. This led to an extremely high false positive rate.

| Model | Reporting Delay | Miss Percentage | FPR | AUC |
|---|---|---|---|---|
| K Means | $-1.58 \pm 10.41$ | 50% | 5% | 0.60 |
| California | $-15 \pm 0$ | 0% | 99% | N/A |

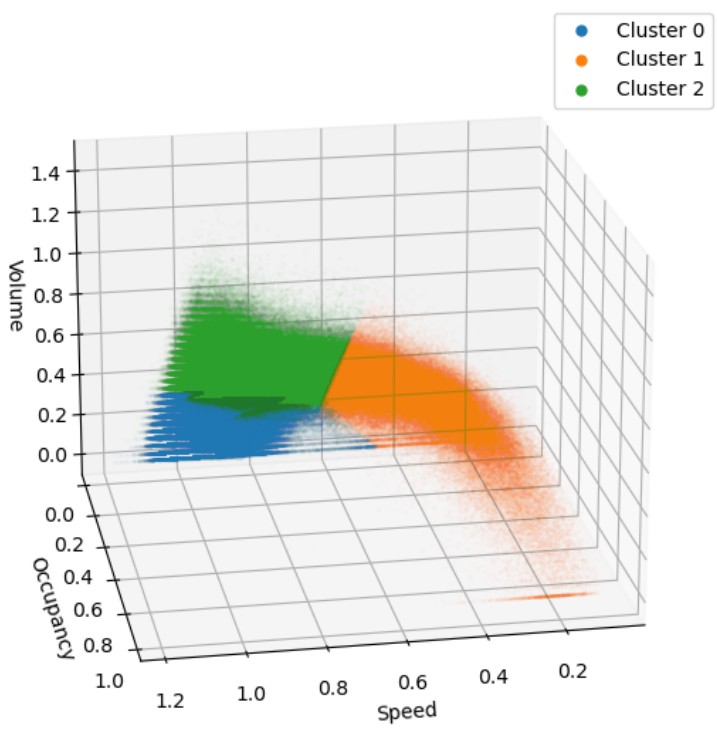

Figure 13: Optimal K Means clustering on the anomaly-free training dataset.

### A.9 Resistance to Impure Data

A common assumption for autoencoder anomaly detection is that the training dataset is completely nominal. To help achieve this assumption, we manually labeled anomalies that were not included in the incident logs. Despite our efforts, it is almost impossible to guarantee our dataset is free of anomalies. Therefore, we ran an additional experiment to determine how important manually removing additional anomalies was.

Table 6: Impact of removing the anomalies that were manually labeled by a human expert. These do not include crashes (all crashes were removed from the autoencoder training data). The threshold was fixed such that the FPR was 5%. Reconstruction error was on anomaly-free validation data. A graph autoencoder (GCN encoder and decoder) was used.

| Training Data | Reporting Delay | Miss Percentage | Recons. MSE | AUC |
|---|---|---|---|---|
| With Anomalies | $-8.45 \pm 7.81$ | **8.3**% | 0.0097 | 0.69 |
| Without Anomalies | $\mathbf{-10.20 \pm 5.98}$ | 25% | **0.0095** | **0.70** |

Table 6 compares the performance of the GCN model with and without additional anomalies in the training set. When the additional anomalies were removed, the reporting delay significantly dropped and the AUC marginally increased. This implies that manually labeling and removing anomalies was an important step. However, the model without anomalies removed missed fewer anomalies. Including some anomalies in the training data led to a more robust model that detected more crashes.

To further explore this phenomenon, we can view the training loss curve in Figure 14. As is shown, when training with anomalies, the loss has large spikes, likely when encountering an anomalous sample. The model is learning to only reconstruct nominal points, treating the anomalies as noise. This is an interesting finding, as it implies that better performance (in terms of robustness of crash

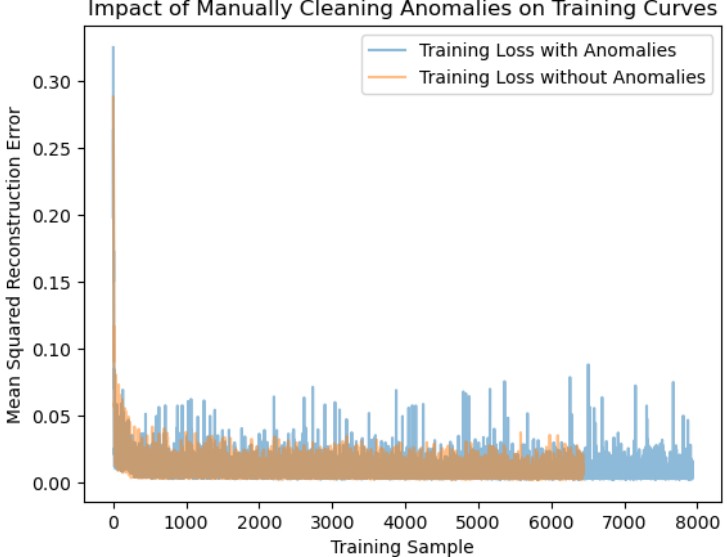

Figure 14: Impact of removing manual anomalies on GCN AE training curve. Notice the training is smoother. Additionally, the loss when training with anomalies spikes. The model is not learning to reconstruct some anomalies, treating them as noise.

detection) might be achieved by balancing the number and types of anomalies in the training dataset. It warrants further research.

## A.10   Data Sampling Rate Impact

In the FT-AED dataset, sensor data was collected every 30 seconds. To determine the impact of having data at this granularity, we downsampled the data to 1-minute, 2-minute, and 5-minute intervals. If crashes were reported between samples, we assumed that they were reported later rather than earlier. Then, we trained the GCN autoencoder using the training process described in the paper (5% fixed FPR). The results of this experiment are shown in Table 7 below.

Table 7: GCN autoencoder performance as the data sampling rate decreases. This gives some intuition on the importance of fast data sampling for anomaly detection.

| Sampling Rate | Reporting Delay | Miss Percentage | AUC |
|---|---|---|---|
| 30 second | $-10.20 \pm 5.98$ | 25% | 0.70 |
| 1 minute | $-9.78 \pm 3.47$ | 25% | 0.72 |
| 2.5 minute | $-9.55 \pm 4.31$ | 25% | 0.62 |
| 5 minute | $-4.75 \pm 5.23$ | 16.7% | 0.72 |

As the sampling speed decreased, the reporting delay increased. The data loses important information about the traffic system that is necessary for detecting the anomaly. Additionally, data is coming in slower, so any impacts caused by anomalous events may not be present until the next sample is taken. We can see this in the difference between the performance when sampling data every 30 seconds and sampling data every 5 minutes. There is over a 5-minute average gap between their reporting delays.

## A.11   Computational Resources

Our network was small (196 nodes) and, aside from the GCN-LSTM, our models were very parallelizable. Therefore, model training and inference did not take significant computational resources. We trained and evaluated our models on a computer with 256 GB of RAM and a 48-core 2.2 GHz. No GPU was required. Since our dataset is 118 MB, most modern computers should be able to run our models without memory issues.

The most computationally intensive part was the hyperparameter optimization. To reduce any negative impacts from increased computational usage, we optimized hyperparameters on a small validation set consisting of only a single day of data. Then, when possible, we trained models only once on the full dataset.

## A.12    Supplementary Data

Additional data can be accessed in the data repository (`https://vu.edu/ft-aed`).

- We added 24-hour sensor data for 1 full week from 11 months from July 2023 to June 2024. We have processed the data in the same way the data from October that comprises the main dataset. We hope this will address concerns with seasonality, as models can be learned or applied to data from the full seasonal spectrum.
- We added raw event logs for all data. These event logs contain additional contextual information, such as the type of incident that was reported.

## A.13    Societal Impacts

We do not foresee any potential negative societal impacts of our work. We aim to enhance freeway traffic anomaly detection and general anomaly detection methods. In cases where the improvement of anomaly detection has negative societal impacts, our work would contribute to those. However, we do not know of any such cases.

