# Supplementary Materials

## 1 Dataset Documentation Card

We organize the dataset documentation as a datasheet [1].

### 1.1 Motivation

**For what purpose was the dataset created? Was there a specific task in mind? Was there a specific gap that needed to be filled? Please provide a description.**
The dataset was created to address the challenge of early and accurate detection of anomalous events on freeways, such as accidents. The specific task in mind was freeway traffic anomaly detection at the lane level, which involves detecting unusual incidents like vehicle accidents, malfunctions, or severe weather conditions that could affect traffic flow. The dataset fills a significant gap by providing a large-scale, lane-level freeway traffic dataset designed explicitly for anomaly detection, addressing the limitations of existing traffic datasets that either lack anomaly information or have low granularity in data collection. This dataset, called FT-AED (Freeway Traffic Anomalous Event Detection), includes traffic data collected from radar detection sensors and official crash reports from the Nashville Traffic Management Center. The dataset aims to facilitate future research in machine learning and traffic management by providing high-fidelity traffic measurements and ground truth anomaly labels, allowing for the development and benchmarking of new anomaly detection models.
**Who created the dataset (e.g., which team, research group) and on behalf of which entity (e.g., company, institution, organization)?**
This dataset was created by Austin Coursey, Junyi Ji, Marcos Quinones-Grueiro, William Barbour, Yuhang Zhang, Tyler Derr, Gautam Biswas, and Daniel B. Work. The authors are researchers affiliated with Vanderbilt University.
**Who funded the creation of the dataset? If there is an associated grant, please provide the name of the grantor and the grant name and number.**
This work was partly funded by the Tennessee Department of Transportation under the grant RES2023-20. The Tennessee State Government assumes no liability for the contents or use thereof.

### 1.2 Composition

**What do the instances that comprise the dataset represent (e.g., documents, photos, people, countries)? Are there multiple types of instances (e.g., movies, users, and ratings; people and interactions between them; nodes and edges)? Please provide a description.**
There are two types of data in our dataset. (1) The traffic flow measurements obtained from radar detection sensors installed along the Interstate 24 corridor in Tennessee. (2) Anomaly labels, including incident data derived from official crash reports provided by the Nashville Traffic Management Center, as well as manually labeled anomalies based on expert analysis of traffic speed profiles.
**How many instances are there in total (of each type, if appropriate)?**
The dataset contains over 3.7 million traffic flow measurement instances. Specifically, these instances are collected every 30 seconds from 49 radar detection sensors deployed along an 18-mile stretch of Interstate 24. The dataset covers every workday in October 2023, during peak morning traffic hours from 4:00 am to 12:00 pm. Additionally, the dataset includes 42 officially reported crash incidents and 19 manually labeled anomalies, making a total of 61 labeled anomaly instances.
**Does the dataset contain all possible instances or is it a sample (not necessarily random) of**

**instances from a larger set? If the dataset is a sample, then what is the larger set? Is the sample representative of the larger set (e.g., geographic coverage)? If so, please describe how this representativeness was validated/verified.**

The dataset is a sample from a larger set of possible freeway traffic data. It focuses on a specific 18-mile stretch of Interstate 24 near Nashville, Tennessee, and captures data for every workday in October 2023 during peak morning traffic hours (4:00 am to 12:00 pm). This sample was chosen to target a high-traffic period where anomalies such as crashes are more likely to occur.

The dataset aims to be representative of typical freeway traffic conditions and anomalies for this specific region and time period. The selection of sensors along this stretch of freeway and the chosen time window were based on the likelihood of capturing relevant traffic anomalies and ensuring a comprehensive dataset for the intended anomaly detection task. The representativeness was validated through expert analysis and comparison with official crash reports from the Nashville Traffic Management Center.

**What data does each instance consist of? "Raw" data (e.g., unprocessed text or images) or features? In either case, please provide a description.**

Each instance in the dataset consists of features: (1) traffic measurements: numerical values for speed, volume, and occupancy measured every 30 seconds. (2)anomaly labels: indicators of anomalous events, such as crashes, based on official reports and expert analysis.

**Is there a label or target associated with each instance? If so, please provide a description.**

Yes, there is a anomaly label associated with each instance. This label indicates whether the instance is part of an anomalous event, such as a crash. The labels are derived from official crash reports and expert analysis of traffic speed profiles.

**Is any information missing from individual instances? If so, please provide a description, explaining why this information is missing (e.g., because it was unavailable).**

Some instances may have missing information due to sensor malfunctions or data transmission errors. These missing values were addressed using imputation methods to ensure the dataset's usability for anomaly detection models.

**Are relationships between individual instances made explicit (e.g., users' movie ratings, social network links)? If so, please describe how these relationships are made explicit.**

No, relationships between individual instances are not made explicit. Each instance is treated independently based on the traffic measurements and anomaly labels for a specific time and location.

**Are there recommended data splits (e.g., training, development/validation, testing)? If so, please provide a description of these splits, explaining the rationale behind them.**

Yes, there are recommended data splits:

1. **Training Set**: Data from the first 14 workdays of October 2023.
2. **Validation Set**: Data from 5 specific workdays used to tune model parameters.
3. **Testing Set**: Data from the remaining workdays, excluding any day with excessive anomalies.

**Are there any errors, sources of noise, or redundancies in the dataset? If so, please provide a description.**

Yes, the dataset may contain sensor errors, reporting delays, and redundancies, which were addressed through data cleaning and imputation.

**Is the dataset self-contained, or does it link to or otherwise rely on external resources (e.g., websites, tweets, other datasets)?**

Yes, it is self-contained.

**Does the dataset contain data that might be considered confidential (e.g., data that is protected by legal privilege or by doctor–patient confidentiality, data that includes the content of individuals' non-public communications)? If so, please provide a description.**

No, the dataset does not contain data that might be considered confidential.

**Does the dataset contain data that, if viewed directly, might be offensive, insulting, threatening, or might otherwise cause anxiety? If so, please describe why.**

No, all our data are numerical.

## 1.3 Collection Process

**How was the data associated with each instance acquired? Was the data directly observable (e.g., raw text, movie ratings), reported by subjects (e.g., survey responses), or indirectly**

**inferred/derived from other data (e.g., part-of-speech tags, model-based guesses for age or language)?**
The data associated with each instance was directly observable, obtained from radar detection sensors and official crash reports.

**What mechanisms or procedures were used to collect the data (e.g., hardware apparatuses or sensors, manual human curation, software programs, software APIs)? How were these mechanisms or procedures validated?**
The data was collected using radar detection sensors and manual human curation of official crash reports. The sensors were validated for accuracy by previous studies, and the crash reports were cross-referenced for reliability, though there may still be missing event reports, posing challenges for future research.

**If the dataset is a sample from a larger set, what was the sampling strategy (e.g., deterministic, probabilistic with specific sampling probabilities)?**
The dataset is a sample from a larger set, collected deterministically to cover every workday in October 2023 during peak morning traffic hours.

**Who was involved in the data collection process (e.g., students, crowdworkers, contractors) and how were they compensated (e.g., how much were crowdworkers paid)?**
The data collection process involved researchers from Vanderbilt University.

**Over what timeframe was the data collected? Does this timeframe match the creation timeframe of the data associated with the instances (e.g., recent crawl of old news articles)? If not, please describe the timeframe in which the data associated with the instances was created.**
The data was collected over the month of October 2023. This timeframe matches the creation timeframe of the data associated with the instances, as the traffic measurements and anomaly labels were recorded in real-time during this period.

**Were any ethical review processes conducted (e.g., by an institutional review board)?**
No, such processes are unnecessary in our case.

## 1.4 Preprocessing/cleaning/labeling

**Was any preprocessing/cleaning/labeling of the data done (e.g., discretization or bucketing, tokenization, part-of-speech tagging, SIFT feature extraction, removal of instances, processing of missing values)? If so, please provide a description.**
Yes, preprocessing, cleaning, and labeling of the data were done:

1. **Preprocessing**: Missing values were imputed using domain-specific methods to enforce wave-like patterns in traffic speeds.
2. **Cleaning**: Sensor data was cleaned to remove inaccuracies and outliers.
3. **Labeling**: Anomaly labels were added based on official crash reports and expert analysis of traffic speed profiles.

**Was the "raw" data saved in addition to the preprocessed/cleaned/labeled data (e.g., to support unanticipated future uses)? If so, please provide a link or other access point to the "raw" data.**
No, only the preprocessed, cleaned, and labeled data was saved to support the intended use of the dataset for anomaly detection.

**Is the software that was used to preprocess/clean/label the data available? If so, please provide a link or other access point.**
No.

## 1.5 Uses

**Has the dataset been used for any tasks already? If so, please provide a description.**
Yes, the dataset has been used for benchmarking various deep learning anomaly detection models. It was employed to evaluate the performance of different autoencoder-based models for the task of lane-level freeway anomaly detection.

**Is there a repository that links to any or all papers or systems that use the dataset? If so, please provide a link or other access point.**
No, but we may create one in the future.

**What (other) tasks could the dataset be used for?**

Incident impact analysis: studying the impact of traffic incidents on overall traffic flow.

**Is there anything about the composition of the dataset or the way it was collected and preprocessed/cleaned/labeled that might impact future uses?**
Yes, the imputation of missing data, manual labeling of anomalies, and limited temporal coverage to weekday mornings in October 2023 might impact future uses.

**Are there tasks for which the dataset should not be used? If so, please provide a description.**
No, users could use our dataset in any task as long as it does not violate laws.

## 1.6 Distribution

**Will the dataset be distributed to third parties outside of the entity (e.g., company, institution, organization) on behalf of which the dataset was created? If so, please provide a description.**
No, it will always be held on GitHub.

**How will the dataset will be distributed (e.g., tarball on website, API, GitHub)? Does the dataset have a digital object identifier (DOI)?**
The dataset is available at: `https://github.com/acoursey3/freeway-anomaly-data`. The dataset does not have a digital object identifier currently, we will release one in the future once approved by Vanderbilt library.

**When will the dataset be distributed?**
On June 12, 2024.

**Will the dataset be distributed under a copyright or other intellectual property (IP) license, and/or under applicable terms of use (ToU)? If so, please describe this license and/or ToU, and provide a link or other access point to.**
Yes, the dataset will be distributed under a Creative Commons Attribution-NonCommercial (CC BY-NC) license. For more information, please visit: `https://github.com/acoursey3/freeway-anomaly-data`.

**Have any third parties imposed IP-based or other restrictions on the data associated with the instances? If so, please describe these restrictions, and provide a link or other access point to, or otherwise reproduce, any relevant licensing terms, as well as any fees associated with these restrictions.**
No, there are no third-party IP-based or other restrictions on the data associated with the instances.

**Do any export controls or other regulatory restrictions apply to the dataset or to individual instances? If so, please describe these restrictions, and provide a link or other access point to, or otherwise reproduce, any supporting documentation.**
No.

## 1.7 Maintenance

**Who will be supporting/hosting/maintaining the dataset?**
The authors of the paper.

**How can the owner/curator/manager of the dataset be contacted (e.g., email address)?**
Please contact this email address: austin.c.coursey@vanderbilt.edu, junyi.ji@vanderbilt.edu.

**Is there an erratum? If so, please provide a link or other access point.**
Users can use GitHub to report issues or bugs.

**Will the dataset be updated (e.g., to correct labeling errors, add new instances, delete instances)? If so, please describe how often, by whom, and how updates will be communicated to dataset consumers (e.g., mailing list, GitHub)?**
Yes, the authors will actively update the code and data on GitHub.

**If the dataset relates to people, are there applicable limits on the retention of the data associated with the instances (e.g., were the individuals in question told that their data would be retained for a fixed period of time and then deleted)? If so, please describe these limits and explain how they will be enforced.**
The dataset does not relate to people.

**Will older versions of the dataset continue to be supported/hosted/maintained? If so, please describe how. If not, please describe how its obsolescence will be communicated to dataset consumers.**
Yes, we will provide the information on GitHub.

**If others want to extend/augment/build on/contribute to the dataset, is there a mechanism for them to do so? If so, please provide a description. Will these contributions be validated/verified?**

**If so, please describe how. If not, why not? Is there a process for communicating/distributing these contributions to dataset consumers? If so, please provide a description.**
Yes, we welcome users to submit pull requests on GitHub, and we will actively validate the requests.

## 2   Additional Supplementary Materials

### 2.1   Accessing the Dataset

The dataset can be accessed directly from the following repository. `https://github.com/acoursey3/freeway-anomaly-data` It is entirely contained in the `nashville_freeway_anomaly.csv` file. It can be read with Python by following the instructions shown in the `Demo.ipynb` file.

### 2.2   Author Statement

The authors of this paper bear full responsibility for any unforeseen violation of rights that may come from the collection of the data included in this research. This dataset is licensed under the CC BY-NC license. See the data repository for more information.

### 2.3   Hosting, Maintenance, and Licensing Plan

Our dataset will continue to be hosted on GitHub (`https://github.com/acoursey3/freeway-anomaly-data`). We do not foresee any updates to the dataset. Dataset users can reach out to the corresponding authors of this paper through email with questions or concerns. If unexpected issues are brought to our attention, the repository will be updated to contain another version of the dataset with an appropriate update message in the `README.md` file.

This dataset is licensed under the CC BY-NC license. See the data repository for more information.

### 2.4   Previous Reviews

A version of this paper was previously reviewed at the International Conference on Knowledge Discovery and Data Mining. The previous version of this paper focused on the machine learning contributions of our graph neural network approaches. The reviews were happy with our dataset but thought the paper lacked a generalizable machine learning contribution. With these helpful comments in mind, we rewrote the entire paper for this more appropriate venue, focusing on the dataset we created. Additionally, we benchmarked more common anomaly detection methods and expanded the results significantly.

## References

[1] GEBRU, T., MORGENSTERN, J., VECCHIONE, B., VAUGHAN, J. W., WALLACH, H., III, H. D., AND CRAWFORD, K. Datasheets for datasets. *Communications of the ACM 64*, 12 (2021), 86–92.