# OpenReview forum: "FT-AED: Benchmark Dataset for Early Freeway Traffic Anomalous Event Detection"
_NeurIPS.cc/2024/Datasets_and_Benchmarks_Track — NeurIPS 2024 Track Datasets and Benchmarks Poster_

### Official Review · Reviewer_CYVt · 2024-07-24

**Rating:** 7
**Confidence:** 4
**Correctness:** The claims made by the paper appear t…
**Clarity:** The paper is overall well-written, wi…

**Review:**

The paper addresses the critical issue of minimizing delays in the detection and reporting of traffic incidents on highways, a problem with significant safety implications. The work presents a large-scale highway traffic dataset curated for anomaly detection at the lane level, collected using physical radar sensors placed along a major US Interstate highway. Anomalies in the dataset were labeled using real incident reports from a local traffic management center and expert analysis techniques such as space-time diagrams. The authors define and tackle a new problem of lane-level anomaly detection, focusing on the minimization of incident reporting delays. Furthermore, they establish a benchmark based on the dataset and problem definition, evaluating several published anomaly detection methods as baselines for future research, demonstrating a notable reduction in anomaly detection times. Nevertheless, I believe exploring the impact of sensor sampling frequency on AE model performance could enhance the study. Despite practical limitations preventing data collection at intervals shorter than 30 seconds, subsampling the data to longer intervals and experimenting with these could provide insights into the influence of sensor frequency on the quality of trained models, especially regarding false negative and reconstruction error metrics. These findings could help determine if upgrading to faster sensors would be beneficial, thereby potentially further enhancing the dataset's advantages and the efficacy of subsequent detection methods. Overall, the paper is of high quality, clear, original, and holds significant potential for improving highway safety through advanced anomaly detection.

**Strengths:**

1. The work focuses on minimizing the delays in the detection and reporting of traffic incidents on highways, which is a problem of great impact and potentially life-saving if solved reliably.
1. The work provides a large-scale highway traffic dataset specifically curated for anomaly detection at the lane levels. The dataset was collected using actual physical radar sensors placed along a major US Interstate highway.
1. The labeling of the anomalies amongst the collected dataset was facilitated using real incident reports from the local traffic management center and expert analysis techniques such as space-time diagrams.
1. Based on the collected and annotated lane-level dataset, the authors define and tackle a new problem of anomaly detection at lane levels and focus on minimizing the delay of the reporting of the incidents.
1. The paper went one step further, establishing a benchmark based on the constructed dataset and lane-level problem and evaluating a series of published anomaly detection methods as baselines for future research. The evaluation demonstrates a notable reduction in anomaly detection.

**Additional Feedback:**

I have no additional comments.

**Documentation:**

Documentations regarding the dataset and codebase are adequately provided in the main paper, the released dataset and codebase READMEs, and the interactive website linked in the paper.

**Ethics:**

There appear to be no notable ethical concerns regarding this work due to the nature of the dataset (radar data vs. image data).

**Limitations:**

The authors have addressed the limitations of their work in great detail in Section 5.

**Opportunities For Improvement:**

Exploring the impact of sensor sampling frequency on the AE model training and testing performance under the proposed metrics could be beneficial. While practical limitations may prevent data collection at intervals shorter than 30 seconds, the authors could still investigate this impact by subsampling the 30-second data into, say, 60-second intervals. By performing experiments on the subsampled data, the authors can assess the influence of sensor frequency on the quality of the trained models gauged by the proposed metrics, especially the false negatives and reconstruction errors. These insights would help determine whether upgrading to sensors with faster sampling rates would be advantageous, thereby potentially further bolstering the benefits of the proposed dataset and any subsequent detection methods developed from it.

**Relation To Prior Work:**

The paper clearly discussed its contributions in relation to the prior work not only in Section 1 but also in great detail in Section 2.

**Summary And Contributions:**

This work focuses on automating and enhancing early and accurate detection of incidents and anomalous events on interstates and highways in order to reduce the risk of secondary accidents and assist in easing traffic congestion. The authors construct a large-scale, lane-level highway dataset for anomaly detection with extensive length, lane information, and sensor measurements. The paper also collected official reports from the local traffic management center to facilitate the labeling of the anomalies in the dataset. Furthermore, the paper demonstrates the benefits of the constructed dataset by establishing a series of metrics, benchmarking a series of ML-based anomaly detection models using the metrics, and then ultimately reducing reporting delays per their evaluations while maintaining reasonable false negative/positive rates.

---

> ### Author Rebuttal · Authors · 2024-08-16
>
> Hello reviewer CYVt. Thank you for your detailed evaluation of our paper. Your feedback will be very helpful in improving the quality of our paper! Please see our response below. In summary, we have run your proposed experiment and analyzed the impact of the sampling rate.
>
> > Nevertheless, I believe exploring the impact of sensor sampling frequency on AE model performance could enhance the study. Despite practical limitations preventing data collection at intervals shorter than 30 seconds, subsampling the data to longer intervals and experimenting with these could provide insights into the influence of sensor frequency on the quality of trained models, especially regarding false negative and reconstruction error metrics. These findings could help determine if upgrading to faster sensors would be beneficial, thereby potentially further enhancing the dataset's advantages and the efficacy of subsequent detection methods.
>
> **Response:** This is a great idea for an experiment to enhance the study. We performed this experiment, and the results of this are shown in the text below which have been included in the paper. Overall, the results hint that we may not need data faster than 30 seconds, as even with 1-minute intervals we obtain pretty strong performance. It also justifies the strengths of obtaining data every 30 seconds instead of every 5 minutes.
>
> *In the FT-AED dataset, sensor data was collected every 30 seconds. To determine the impact of having data at this granularity, we downsampled the data to 1-minute, 2-minute, and 5-minute intervals. If crashes were reported between samples, we assumed that they were reported later rather than earlier. Then, we trained the GCN autoencoder using the training process described in the paper (5\% fixed FPR). The results of this experiment are shown in table below.*
>
> | Sampling Rate | Reporting Delay   | Miss Percentage | AUC   |
> |---------------|-------------------|-----------------|-------|
> | 30 second     | $-10.20\pm5.98$   | 25%             | 0.70  |
> | 1 minute      | $-9.78\pm3.47$    | 25%             | 0.72  |
> | 2.5 minute    | $-9.55\pm4.31$    | 25%             | 0.62  |
> | 5 minute      | $-4.75\pm5.23$    | 16.7%           | 0.72  |
>
> *As the sampling speed decreased, the reporting delay increased. The data loses important information about the traffic system that is necessary for detecting the anomaly. Additionally, data is coming in slower, so any impacts caused by anomalous events may not be present until the next sample is taken. We can see this in the difference between the performance when sampling data every 30 seconds and sampling data every 5 minutes. There is over a 5-minute average gap between their reporting delays.*

---

> > ### Comment · Area_Chair_tVE6 · 2024-08-29
> >
> > Can Reviewer CYVt please comment on the authors' rebuttal? Thank you.

---

> > ### Comment · Reviewer_CYVt · 2024-08-29
> >
> > Thank you for the additional experiments! I am glad to see that the reporting delay is indeed inversely proportional to the sampling rate, and that the reporting delay decays exponentially, indicating that a 30-second sampling rate may suffice for the application and any higher sampling rate may not yield any cost-effective benefits. Thanks again!

---

### Official Review · Reviewer_z8qJ · 2024-07-24
**Reviews**

**Rating:** 6
**Confidence:** 4
**Correctness:** Yes
**Clarity:** Yes

**Review:**

This work introduces the FT-AED benchmark dataset, which is the first large-scale lane-level freeway traffic dataset designed specifically for anomaly detection, addressing a critical gap in the existing datasets. The dataset captures real-world challenges such as delayed incident reporting, which is an important consideration for developing practical anomaly detection solutions. The authors conducted a thorough benchmarking of various deep learning anomaly detection methods, establishing a baseline for the novel task of lane-level freeway anomaly detection. They provided a detailed evaluation of the benchmarked models, including their ability to reduce reporting delays and detect a high percentage of crashes.

While the dataset offers valuable insights, it also has some limitations. The dataset only covers a single month of data, which may limit the ability to capture seasonal or long-term trends in freeway traffic anomalies. Expanding the temporal scope of the dataset could provide a more comprehensive understanding of the problem. Additionally, the dataset only includes traffic sensor data and manually labeled anomalies, but lacks other potentially relevant contextual information. Incorporating additional contextual data could enhance the ability to understand the root causes and triggers of freeway anomalies. The authors also acknowledge that the manual labeling of anomalies may have introduced some biases or inconsistencies, which could affect the reliability of the ground truth information. Exploring techniques to improve the accuracy and consistency of the labeling process could enhance the dataset's quality.

**Strengths:**

This is the first large-scale lane-level freeway traffic dataset designed specifically for anomaly detection, addressing a critical gap in the existing datasets. The dataset captures real-world challenges such as delayed incident reporting, which is an important consideration for developing practical anomaly detection solutions.

The authors conducted a thorough benchmarking of various deep learning anomaly detection methods, establishing a baseline for the novel task of lane-level freeway anomaly detection.

The authors provided a thorough evaluation of the benchmarked anomaly detection models, including their ability to reduce reporting delays and detect a high percentage of crashes.

**Additional Feedback:**

Please refer to my identified weak points for further improvement.

**Documentation:**

Yes

**Limitations:**

Yes

**Opportunities For Improvement:**

The dataset only covers a single month of data, which may limit the ability to capture seasonal or long-term trends in freeway traffic anomalies. Expanding the temporal scope of the dataset, such as including data from multiple seasons or years, could provide a more comprehensive understanding of the problem.

The dataset only includes traffic sensor data and manually labeled anomalies, but lacks other potentially relevant contextual information, such as weather conditions, event schedules, or road construction activities. Incorporating additional contextual data could enhance the ability to understand the root causes and triggers of freeway anomalies.

The authors acknowledge that the manual labeling of anomalies may have introduced some biases or inconsistencies, which could affect the reliability of the ground truth information. Exploring techniques to improve the accuracy and consistency of the labeling process, such as utilizing multiple annotators or automated methods, could enhance the dataset's quality.

**Relation To Prior Work:**

Yes

**Summary And Contributions:**

This work introduces the FT-AED benchmark dataset, which is the first large-scale lane-level freeway traffic dataset designed for anomaly detection, such as accidents and incidents. The dataset consists of over 3.7 million sensor measurements collected every 30 seconds from 49 radar detection sensors along an 18-mile stretch of Interstate 24 in Tennessee. The authors also manually labeled all anomalies in the dataset, including crashes reported by the Nashville Traffic Management Center.

---

> ### Author Rebuttal · Authors · 2024-08-16
>
> Hello reviewer z8qJ. Thank you for taking the time to provide us such a thoughtful review. We greatly appreciate your feedback and believe it will improve the quality of our paper. Please see our responses to your comments below. In summary, we added another 11 months of data with 1 full week from each month, added the raw event logs as additional context, and discussed concerns with biases from human labeling.
>
> >The dataset only covers a single month of data, which may limit the ability to capture seasonal or long-term trends in freeway traffic anomalies. Expanding the temporal scope of the dataset could provide a more comprehensive understanding of the problem.
>
> **Response:** Thank you for this suggestion! We have added another 11 months of data with 1 full week for each month. We have also added the raw event logs corresponding to this data. This is posted on the data repository (https://github.com/acoursey3/freeway-anomaly-data/) under the Supplementary Data section of the README. We are still working on processing and gathering the rest of this data for release, including processing the event logs which are very time-consuming to parse. This additional data should help alleviate concerns of limited generality over seasonality or weather conditions!
>
> > Additionally, the dataset only includes traffic sensor data and manually labeled anomalies, but lacks other potentially relevant contextual information. Incorporating additional contextual data could enhance the ability to understand the root causes and triggers of freeway anomalies.
>
> **Response:** This is absolutely true. Additional contextual information would only serve to improve the training and evaluation of these models. Understanding the root causes of anomalous events is a difficult challenge and may not always be possible. To address these concerns, we added the raw event logs that may contain additional contextual information. This raw log contains a column called “classification” that labels each event (e.g., road construction, abandoned vehicle, etc.) In the original dataset, we processed these to determine the times of crashes.
>
> > The authors also acknowledge that the manual labeling of anomalies may have introduced some biases or inconsistencies, which could affect the reliability of the ground truth information. Exploring techniques to improve the accuracy and consistency of the labeling process could enhance the dataset's quality.
>
> **Response:** This is a reasonable concern. To mitigate the impacts of this concern, we only used true labels for evaluation. These ground truth labels came directly from crash report logs, so they accurately represent when a crash was reported. We also explored the impact of missing anomalies on the training process (Appendix A.8). To address this concern for model validation, we manually adjusted the reporting labels for all crashes and added this as supplementary data. With this, researchers can determine how close to a non-delayed reporting the model detects. It is important to note that the delayed labels should still primarily be used because they reflect how well the model addresses the real-world challenge of automatically detecting crashes before humans report them. We also included the raw event logs as contextual data and support for the labeling.

---

### Official Review · Reviewer_DJ9r · 2024-07-24
**A good supplement to the benchmark dataset for freeway traffic anomalous event detection task**

**Rating:** 7
**Confidence:** 5

**Review:**

Pros:
1.	Quality: he overall quality of this paper is relatively high. The usability and generalizability of the collected data was ensured in various ways such as domain-specific imputation. The paper also conducts a number of experiments to demonstrate the usability of the data and compares it with many existing methods.
2.	Clarity: The paper is well-structured and clearly explains the dataset creation process and experiments.
3.	Originality: The FT-AED dataset is the first of its kind, offering lane-level granularity and addressing the crucial issue of delayed crash reporting in freeway traffic anomaly detection.
4.	Significance: This work has potential for substantial real-world impact, as improved anomaly detection could lead to reduce the risk of secondary accidents and ease traffic congestion.
Cons:
1.	Limited timeframe: The dataset covers only one month of data, which may not capture seasonal variations or long-term traffic patterns.
2.	Geographical limitation: The data is collected from a single stretch of highway near Nashville, potentially limiting its generalizability to other regions or traffic conditions.
3.	Lack of comparison methods: the paper compares a variety of autoencoder-based methods and lacks a comparison of some other anomaly detection techniques.
4.	Discussion of evaluation metrics: due to the small amount of data, it is difficult to directly use Miss Percentage for comparison, it is better to discuss whether there are better evaluation metrics.

**Strengths:**

Please Refer to the Review part.

**Additional Feedback:**

N/A. Please refer to the Review and Opportunities For Improvement section.

**Clarity:**

The paper is well-structured and clearly explains the dataset creation process and experiments.

**Correctness:**

The claims made in the paper appear to be correct, and the dataset and benchmark are constructed and evaluated in a sound manner.

**Documentation:**

The documentation for the FT-AED dataset and benchmark appears to be comprehensive and well-detailed.

**Limitations:**

The authors have made some efforts to analysis address the limitations. They have dedicated Section 5 to discussing future directions and limitations, which shows a proactive approach to acknowledging the constraints of their research.

**Opportunities For Improvement:**

1 more introduction of some other anomaly detection techniques to judge their effectiveness on the proposed dataset.
2 discuss the usage scenarios of different evaluation metrics to enhance generalizability.

**Relation To Prior Work:**

Yes, the paper clearly shows the differentiation from prior works in several key aspects, including dataset specialties, finer granularity with lane-level information, etc.

**Summary And Contributions:**

This paper introduces the Freeway Traffic Anomalous Event Detection (FT-AED) dataset, the first large-scale lane-level freeway traffic dataset designed specifically for anomaly detection. The dataset comprises over 3.7 million sensor measurements collected from 49 radar detection sensors with 42 official crash reports.

---

> ### Author Rebuttal · Authors · 2024-08-16
>
> Hello reviewer DJ9r. Thank you for your well-thought-out review of our paper. Your detailed feedback will be valuable in shaping the direction of our paper. Your valuable comments will significantly improve the quality of our paper. We have tried to address all the concerns. As a summary, we added another 11 months of data with 1 full week for each month, we added raw event logs to help contextualize anomalies, and we added two more non-reconstruction-based baseline approaches.
>
> Please see our detailed responses below.
>
> > 1. Limited timeframe: The dataset covers only one month of data, which may not capture seasonal variations or long-term traffic patterns.
>
> **Response:** Thank you for bringing up this point! We have added another 11 months of data with 1 full week for each month. We have also added the raw event logs corresponding to this data. This is posted on the data repository (https://github.com/acoursey3/freeway-anomaly-data/) under the Supplementary Data section of the README. We are still working on processing and gathering the rest of this data for release, including processing the event logs which are very time-consuming to parse. This additional data should help alleviate concerns of limited generality over seasonality or weather conditions!
>
> > 2. Geographical limitation: The data is collected from a single stretch of highway near Nashville, potentially limiting its generalizability to other regions or traffic conditions.
>
> **Response:** We agree on this point and it should be considered as a limitation of our dataset. Dataset users should be aware that models developed based on it may not generalize to new geographic locations. Before applying in the real world, models should undergo careful inspection to determine whether they are appropriate for the location they are in. As the largest dataset for lane-level freeway traffic anomaly detection, we think the dataset can benefit the community.
>
> > 3. Lack of comparison methods: the paper compares a variety of autoencoder-based methods and lacks a comparison of some other anomaly detection techniques.
>
> **Response:** Thank you for your suggestion! A major goal in releasing this dataset is to facilitate future development of techniques for freeway traffic anomalous event detection. Autoencoder techniques are well-suited to unsupervised anomaly detection tasks. They do not have to rely on the accuracy of event labels. To address concerns about the diversity of the baselines, we added two more baselines, which will be discussed in the paper. The first is a clustering approach using KMeans. We optimized the number of clusters such that the inertia (within-cluster sum-of-squares) of the anomaly-free training data converged. The second is a popular rule-based approach called the California algorithm [1]. We started with thresholds from the original paper [1] and ran an optimization script to try to achieve a 5% false positive rate, but this method still led to a node being detected as anomalous at almost every point in time. Instead of a single threshold to tune, this method requires 3. (Note that this method is known for being extremely difficult to tune [2], highlighting the benefits of learning-based approaches.) The results of these can be seen in the table below.
>
> | Model       | Reporting Delay  | Miss Percentage | FPR  | AUC  |
> |-------------|------------------|-----------------|------|------|
> | K Means     | -1.58 ± 10.41    | 50%             | 5%   | 0.60 |
> | California  | -15 ± 0          | 0%              | 99%  | N/A  |
>
> **Table**: Performance of additional anomaly detection methods. Note that the California algorithm almost always detects at least one node as an anomaly, leading to an extremely high false positive rate.
>
> > 4. Discussion of evaluation metrics: due to the small amount of data, it is difficult to directly use Miss Percentage for comparison, it is better to discuss whether there are better evaluation metrics.
>
> **Response:** Indeed, miss percentage on its own is not the most informative evaluation metric, especially due to the small number of labeled crashes. However, we also compute and present the reduction in reporting delay, reconstruction error, and AUC. The reduction in reporting delay is the main metric of interest, as it demonstrates how much faster an anomaly detection algorithm can flag a crash than be reported by traffic management. The validation reconstruction error quantifies how accurately the model has learned to reconstruct unseen anomaly-free points. The AUC quantifies the overall node-level anomaly detection abilities of the model, considering the tradeoff between false positive and true positive rate at various thresholds. We believe that all these metrics are important for wholistically comparing anomaly detection methods on our dataset. These metrics do have some limitations, primarily coming from the ambiguity of when an anomaly begins or ends. We hope that future research will focus on addressing this challenging problem by developing more metrics.
>
> To help alleviate these concerns, **we have added the raw event logs that contain more contextual information about crashes**. These could be used for the development of future evaluation metrics.
>
> [1] Payne, Howard J., and Samuel C. Tignor. "Freeway incident-detection algorithms based on decision trees with states." Transportation Research Record 682 (1978).
> [2] MARTIN, P. T., PERRIN, J., HANSEN, B., KUMP, R., MOORE, D., ET AL. Incident detection
> algorithm evaluation. Tech. rep., Upper Great Plains Transportation Institute, 2001.

---

> > ### Comment · Area_Chair_tVE6 · 2024-08-29
> >
> > Can Reviewer DJ9r please comment on the authors' rebuttal? Thank you.

---

### Official Review · Reviewer_DxHH · 2024-07-27
**FT-AED: dataset benchmark for reducing reporting times for anomalous traffic events**

**Rating:** 7
**Confidence:** 3

**Review:**

The paper is well written, the problem the work addresses is clearly introduced, as well as the distinctions between the current dataset and other datasets for traffic analysis, mainly temporal frequency, data modality and annotations specific for tasks of anomaly detection. The dataset contains true labels from event logs, as well as anomalies annotated using expert knowledge that mitigate potential missing reports from the official logs.

Pros:
* The collection of a traffic analysis dataset with characteristics that make it suitable for anomaly detection, as well as the introduction of the task of lane-level anomaly detection, with the graph representation of data accounts for both temporal and spatial dimensions
* The initial experiments, although focusing solely on autoencoder reconstructions, reveal promising results, detecting anomalies within a considerable timespan to a manual report, showing potential in using anomaly detection methods for reducing subsequent traffic problems and potentially avoiding follow-up accidents in the area.

Cons:
* the set of baselines needs to be diversified, as all of the methods are reconstruction-based variants
* punctual anomalies might induce long-lasting traffic conditions that might result in connected identifications of anomalous events that would be difficult to distinguish from the root cause

**Strengths:**

* The newly introduced dataset provides real world radar data at a rate of 30 seconds per node, a frequency that makes it attractive for traffic anomaly detection study compared to the currently existing datasets.
* The initial experiments evaluate several types of autoencoder baselines (GNNs, Transformers and MLPs), as well as experimenting with the temporal aspect of the data at a both model and framework level.
* Data characteristic and temporal frequency allow the computation of reduction in reporting delays, a highly important metric in the context of automated traffic anomaly detection.

**Additional Feedback:**

It seems that the Transformer Auto-Encoder model achieves the worst reporting delays (with detection time coming later than the manual report, which is often delayed in the order of minutes after the occurrence). Is there any insight on this poor performance? I find this surprising, as it is mentioned that the transformer method uses temporal data, using a time window of recent points.

**Clarity:**

The paper is well written and there are sufficient details on the problem introduction, relation to prior work, dataset construction and experimental setup.

**Correctness:**

The claims seem to be correct and the dataset constructed in a sound way. The benchmark result use adequate evaluation methods (with due limitations discussed in the paper) and a reasonably sized number of baselines.

**Documentation:**

The dataset is made openly available. The authors also included an example Jupyter notebook in their GitHub repository. Details on dataset collections and properties are also adequately discussed in the paper.

**Ethics:**

There are no ethical concerns with the submission.

**Limitations:**

The authors dedicate a section for discussions on future works and current limitations, mentioning training at a lane-level as opposed to validation, necessity for more adequate performance metrics than AUC, need for better methods of regularizing factors for the reconstructions of the speed without sacrificing the autoencoder bottleneck and need for more anomaly detection approaches.

**Opportunities For Improvement:**

* The experimental setup focuses exclusively on methods based on autoencoder reconstructions. While initial result show promising results, other types of approaches should be investigated, as reconstruction-based methods might lose some of the subtle patterns present in data.
* As some of the methods excel in reducing the report delay, with others obtaining lower miss percentages or reconstruction errors, it would be interesting to observe the effect of combining / ensembling strategies.
* Seasonal phenomena might induce strong forms of out-of-distribution measurements compared to the collected set. For instance, meteo conditions in winter or foggy seasons might result in different traffic conditions compared to the reference dataset. It would be useful to measure the robustness of anomaly detection methods to such conditions, if data extension is possible..

**Relation To Prior Work:**

The authors discuss the existence of large datasets for traffic forecasting and propose the new dataset as an evaluation benchmark for freeway lane-level anomalous event detection, introducing a novel problem in the field of traffic analysis. The new dataset consists of sensor measurement, which makes it easier to annotate and deploy as opposed to video-based solutions. The authors note that existing sensor datasets contain no anomaly information or the anomaly labels have a suboptimal granularity.

**Summary And Contributions:**

The authors release the Freeway Traffic Anomalous Event Detection (FT-AED) dataset as a benchmark dataset for the novel problem of lane-level anomaly detection in freeway monitoring. The dataset consists of data collected every 30 seconds over the weekdays in a month, comprising over 3.7 million sensor measurements. The sensors collect speed, occupancy and traffic volume data and the problem is formulated as a node-level graph anomaly detection.

The present work provides an initial evaluation of several auto-encoder methods on the proposed task, including graph-based autoencoder methods, Transformer autoencoders and MLP autoencoders. The metrics of interest are AUC, reconstruction MSE, miss percentage, as well as Reporting Delay, defined as the difference between the timestamp of detection and the timestamp of incident reporting, important in assessing the performance of an anoamly detection system in the context of reducing the delays provided by manual reporting, and potentially reducing the risk of secondary, follow-up accidents.

Initial experiments show promising results, with baseline methods reducing the report delays by over 10 minutes on average, while successfully detecting 75% of the crashes. On a specific example analysed as a case study, an accident is detected as an anomaly within am impressive time of 17 minutes before it is officially reported. The results support the utility of the FT-AED benchmark dataset in further developing lane-level anomaly detection methods that substantially outperform manual reports and potentially prevent further traffic problems.

---

> ### Author Rebuttal · Authors · 2024-08-16
>
> Hello reviewer DxHH. Thank you for taking the time to provide us with such detailed feedback. Your valuable comments will significantly improve the quality of our paper. We have tried to address all the concerns. As a summary of our major changes, we added two more non-reconstruction-based baselines and we added more data (1 full week from 11 more months) to address seasonality concerns.
>
> Our detailed responses to your comments are attached below.
>
> >the set of baselines needs to be diversified, as all of the methods are reconstruction-based variants
>
> **Response:** Thank you for your suggestion! A major goal in releasing this dataset is to facilitate future development of techniques for freeway traffic anomalous event detection. Autoencoder techniques are well-suited to unsupervised anomaly detection tasks. They do not have to rely on the accuracy of event labels. To address concerns with the diversity of the baselines, we added two more baselines, which will be discussed in the paper. The first is a clustering approach using KMeans. We optimized the number of clusters such that the inertia (within-cluster sum-of-squares) of the anomaly-free training data converged. The second is a popular rule-based approach called the California algorithm [1]. We started with thresholds from the original paper [1] and ran an optimization script to try to achieve a 5% false positive rate, but this method still led to a node being detected as anomalous at almost every point in time. Instead of a single threshold to tune, this method requires 3. (Note that this method is known for being extremely difficult to tune [2], highlighting the benefits of learning-based approaches.) The results of these can be seen in the table below.
>
> | Model       | Reporting Delay  | Miss Percentage | FPR  | AUC  |
> |-------------|------------------|-----------------|------|------|
> | K Means     | -1.58 ± 10.41    | 50%             | 5%   | 0.60 |
> | California  | -15 ± 0          | 0%              | 99%  | N/A  |
>
> >punctual anomalies might induce long-lasting traffic conditions that might result in connected identifications of anomalous events that would be difficult to distinguish from the root cause
>
> **Response:** This is certainly a challenge of this dataset. Unfortunately, this is an inherent limitation in the data source and system at hand. The main way to address this would be with improved ground truth incident detection and reporting.
>
> >reconstruction-based methods might lose some of the subtle patterns present in data.
>
> **Response:** This point is an interesting one. Reconstruction-based methods lose information by design. Qualitatively, we observed that the speed reconstructions lost some of the “traffic waves” present before reconstruction. We have been actively researching this topic since the submission of this paper. Having an autoencoder fail to reconstruct anomalous behavior but still maintain the dynamics of the system is an interesting research challenge, and approaches such as physics-informed neural networks may be beneficial (i.e., penalize the loss for failing to reconstruct and for not maintaining the wave-like dynamics of the traffic system).
>
> >As some of the methods excel in reducing the report delay, with others obtaining lower miss percentages or reconstruction errors, it would be interesting to observe the effect of combining / ensembling strategies.
>
> **Response:** This is a great idea. This could reduce the number of false positives. While we will leave an in-depth study into this question for future research, we plan to integrate this idea into our real-world deployment of our anomaly detection algorithms.
>
> >Seasonal phenomena might induce strong forms of out-of-distribution measurements compared to the collected set
>
> **Response:** We have released another 11 months of sensor data (1 full week from each month) on the data repository (https://github.com/acoursey3/freeway-anomaly-data/). This should allow for learning more diverse seasonal representations. We included the raw event logs in the data repository and are working on processing them further.
>
> Including weather data is a great idea. However, our sensors do not collect weather information. Many free daily or hourly weather sources are available. We encourage future researchers to use this data corresponding to the provided timestamps and locations to enhance their models.
>
> >It seems that the Transformer Auto-Encoder model achieves the worst reporting delays (with detection time coming later than the manual report, which is often delayed in the order of minutes after the occurrence). Is there any insight on this poor performance?
>
> **Response:** This is a good catch! It’s briefly discussed in the paper on page 8. To further elaborate, a time series of nodes is fed to the transformer. Each node has a 3-dimensional feature space (speed, occupancy, volume). To enforce an autoencoder bottleneck, the latent space must be 1 or 2 dimensional. We suspect that such a small latent space leads to a significant loss of information that causes poor performance. Additionally, the transformer cannot leverage spatial relationships like the graph networks. Without a small latent space, the autoencoder can easily learn the identify function, perfectly reconstructing everything. To avoid this, each node could be treated as a separate feature, leading to 196x3 features. However, this would mean the transformer only detects the time of an anomaly, not the location like the other methods which would not be a fair comparison. The transformer was shown to highlight the importance of both spatial and temporal relationships.
>
> [1] Payne, Howard J., and Samuel C. Tignor. "Freeway incident-detection algorithms based on decision trees with states." Transportation Research Record 682 (1978).
> [2] MARTIN, P. T., PERRIN, J., HANSEN, B., KUMP, R., MOORE, D., ET AL. Incident detection
> algorithm evaluation. Tech. rep., Upper Great Plains Transportation Institute, 2001.

---

> > ### Comment · Reviewer_DxHH · 2024-08-28
> >
> > Thank you for the response and for the interesting additional information!

---

### Official Review · Reviewer_LEDx · 2024-07-28
**Review of "FT-AED"**

**Rating:** 6
**Confidence:** 2
**Correctness:** It seems to be correct.
**Clarity:** Yes

**Review:**

Cons:
1. The dataset is collected from a specific geographic location (Interstate 24 toward Nashville, Tennessee), which might limit the generalizability of the findings.

2. The dataset includes 42 crashes identified from the Nashville Traffic Management Center's logs. These logs typically report crashes with a delay due to the time it takes for humans to report. The authors did not use human input  to revise these timestamps. However, they introduced an additional objective to detect crashes as early as possible, even before they are officially recorded in the logs.

3. Since the dataset relies on radar sensor data, identifying false positives can be more challenging and time-consuming for a human compared to when the data is a video stream. This is because video streams provide visual context that aids quicker and more intuitive verification of incidents.

**Strengths:**

1. The FT-AED dataset is the first large-scale, lane-level, freeway traffic dataset designed specifically for anomaly detection. This dataset's granularity, with data collected every 30 seconds across multiple lanes, allows for detailed analysis and modeling.

2. The authors conducted thorough benchmarking of various deep learning models, particularly focusing on autoencoder-based approaches for anomaly detection. This provides a foundational baseline for future research.

3.  The models developed and tested demonstrated the potential to significantly reduce the delay in crash reporting, which is crucial for improving emergency response times and reducing the likelihood of secondary accidents.

**Additional Feedback:**

The dataset can be improved if the timestamps for the official crashes (taken from Nashville Traffic Management Center's logs) are updated. This update can be done based on human feedback, similar to what was done in the article for detecting additional anomalies.
Overall, I lean towards accepting this work because FT-AED is the first large-scale, freeway traffic dataset designed specifically for anomaly detection.

**Documentation:**

No concern from this point of view.

**Limitations:**

The article shoud include a discussion through the lens of interpretability about the advantages and disadvantages of using radar sensor data rather than video streams.

**Opportunities For Improvement:**

The dataset can be improved if the timestamps for the official crashes (taken from Nashville Traffic Management Center's logs) are updated. This update can be done based on human feedback, similar to what was done in the article for detecting additional anomalies.

**Relation To Prior Work:**

Yes

**Summary And Contributions:**

The article introduces the Freeway Traffic Anomalous Event Detection (FT-AED) dataset, which focuses on detecting traffic anomalies such as crashes on freeways. This dataset, collected along Interstate 24 near Nashville, Tennessee, includes over 3.7 million radar sensor measurements across four lanes. The article benchmarks various autoencoder-based deep learning models to demonstrate the dataset's utility for anomaly detection. It is also shown that graph neural network autoencoders were the most effective in detecting the anomalies.

---

> ### Author Rebuttal · Authors · 2024-08-16
>
> Hello reviewer LEDx. Thank you for your careful review of our paper. Your feedback and comments raise good points and are valuable in shaping the revisions of our paper. We have tried to address all the concerns. As a summary, we clarified the geographical limitations, manually adjusted the timestamps for official crashes, and discussed the tradeoff between radar sensor and video data in terms of interpretability. We also included more sensor data in the dataset, with 1 full week per month from another 11 months with 24 hours per day.
>
> Our detailed responses to your comments are attached below.
>
> >The dataset is collected from a specific geographic location (Interstate 24 toward Nashville, Tennessee), which might limit the generalizability of the findings.
>
> This is a valid limitation of the work. Models developed on this dataset will need to be validated on other locations before they can confidently be applied and be considered general. Our data is limited to I-24, but we hope that other communities will be able to eventually release similar datasets to support geographic generalizability of the findings. Open data sources exist for other geographic locations (like PeMS for California), but these data sources have limitations, as discussed in Section 2.1.
>
> >The dataset includes 42 crashes identified from the Nashville Traffic Management Center's logs. These logs typically report crashes with a delay due to the time it takes for humans to report. The authors did not use human input to revise these timestamps.
>
> This is a great point, and it highlights an important challenge in this domain. The reality is that it’s challenging for us to know exactly when an event occurred. The impact of this on the training may be minimal (see Appendix A.8). The natural delay lends itself really well to unsupervised methods, which is why many of our methods were autoencoders. However, adjusting the labels could reduce the conservatism in determining false positives for model evaluation. **With this in mind, we manually adjusted the delayed crash reports based on observations from the space-time diagram. This is released as another feature in the dataset.** This will allow for better assessment into how close detected anomalies are to when real crashes occurred.
>
> We want to emphasize that the reduction in reporting delay will not be heavily impacted by this change (we could make true positives slightly less conservative at the beginning, but we do not know how long “anomalies” last after a crash occurs). This is the metric we really care about since it highlights the utility of these models to traffic management centers. If the reduction in reporting delay is -10 minutes, that means an automated incident detection approach is detecting incidents 10 minutes faster than humans report them. Therefore, we encourage users of the dataset to focus on this metric.
>
> >Since the dataset relies on radar sensor data, identifying false positives can be more challenging and time-consuming for a human compared to when the data is a video stream. This is because video streams provide visual context that aids quicker and more intuitive verification of incidents.
>
> We agree that analyzing a video stream would be a great way to identify false positives. Gathering and processing video data comes with a large set of challenges such as deploying enough cameras to cover all 17 miles of the interstate, privacy concerns, and increased computational load. To help reduce the impact of false positives in our radar data, our quantitative evaluations are only on reported crashes, as we know those are anomalies. Verifying another anomaly outside of this evaluation set is very difficult, and is an inherent limitation in the sensing data. Hopefully future work can address this with multimodal data collection or more rigorous incident reporting.
>
> >The article should include a discussion through the lens of interpretability about the advantages and disadvantages of using radar sensor data rather than video streams
>
> We have added the following text to the background section.
>
> Radar sensors offer advantages over video streams for freeway anomaly detection, particularly regarding privacy, spatial coverage, and real-time data processing. Unlike video cameras, which capture identifiable details and raise privacy concerns, radar data is inherently anonymized, focusing solely on traffic measurements without personal information. Radar sensors also provide more extensive spatial coverage, enabling denser and broader observation. Moreover, radar streams structured data in real-time, making it easier to process and interpret compared to the high-dimensional, unstructured nature of video streams. This makes radar a more robust, efficient, and privacy-friendly solution for large-scale traffic monitoring. The advantage of video streams may come from increased interpretability of model outputs. Instead of requiring domain knowledge or other contextual information to determine whether an anomaly was correctly detected as in radar data, in video data, we can simply view the video. Video streams can serve as a valuable complement to our proposed approach by providing additional detail for refining anomalies detected by radar, especially with nearby cameras.

---

> > ### Comment · Area_Chair_tVE6 · 2024-08-29
> >
> > Can Reviewer LEDx please comment on the authors' rebuttal? Thank you.

---

> > > ### Comment · Reviewer_LEDx · 2024-08-30
> > > **Rebuttal answer**
> > >
> > > I appreciate the authors' rebuttal and their detailed responses. After reviewing their answers, I have decided to maintain my initial score, as they have addressed all of my concerns.

---

### Official Review · Reviewer_9FVR · 2024-07-28

**Rating:** 7
**Confidence:** 4
**Correctness:** Correct.
**Clarity:** The paper is well written.

**Review:**

Please see the strengths and opportunities for improvement.

**Strengths:**

- The paper introduces the FT-AED dataset, the first large-scale lane-level freeway traffic dataset specifically designed for anomaly detection. This dataset includes detailed traffic data captured at high temporal resolution (every 30 seconds), offering a valuable resource for researchers to study and improve traffic anomaly detection methods.

- The authors benchmark a variety of deep learning models, including graph neural network autoencoders, to evaluate their effectiveness in detecting freeway anomalies. Moreover, the paper publicly releases the dataset and preprocessing code,

**Additional Feedback:**

None

**Documentation:**

Yes.

**Ethics:**

No.

**Limitations:**

This paper has discussed its limitations.

**Opportunities For Improvement:**

- While the dataset focuses on freeway traffic anomalies, it primarily includes data for weekday mornings and may not capture a wide range of anomaly types or traffic conditions. It would be better to discuss the inclusion of other anomalies.

- The paper acknowledges challenges with labeling anomalies due to delays and potential inaccuracies in manual incident reporting. The methods used for labeling and evaluating the models, such as assuming a fixed delay for incident reporting, may not accurately reflect real-world conditions. It would be better to discuss further approaches to address the issues caused by delays.

**Relation To Prior Work:**

The relation to prior work is well discussed.

**Summary And Contributions:**

The paper presents a new dataset and benchmarking methods for detecting anomalous events, such as accidents, on freeway systems. The research introduces a large-scale, lane-level dataset that utilizes radar detection sensor data collected along Interstate 24 near Nashville, Tennessee. This dataset includes over 3.7 million sensor measurements and incorporates official crash reports and manually labeled anomalies.
- This work contributes a new data set. It is the first of its kind designed specifically for lane-level freeway anomaly detection. It includes detailed data from 49 radar sensors capturing traffic speed, volume, and occupancy every 30 seconds across four lanes over a month of weekday mornings.
- The paper experiments with various deep learning models, including unsupervised graph neural network autoencoders, to evaluate their effectiveness in detecting traffic anomalies.
- The models tested in the paper demonstrated the potential to significantly reduce the time delay in reporting incidents, detecting 75% of crashes on average 10 minutes before they were officially reported.

---

> ### Author Rebuttal · Authors · 2024-08-16
>
> Hello reviewer 9FVR. Thank you for your detailed review and feedback on our paper! We have tried to address all your concerns. As a summary, we included more sensor data to the dataset, with 1 full week per month from another 11 months with 24 hours per day. We also clarified concerns with the delay in incident reporting.
>
> Our detailed responses to your comments are included below.
>
> >While the dataset focuses on freeway traffic anomalies, it primarily includes data for weekday mornings and may not capture a wide range of anomaly types or traffic conditions. It would be better to discuss the inclusion of other anomalies.
>
> **Response:** The data was limited to weekday mornings because morning peak hour traffic is the primary time of interest to the traffic management center. You are absolutely correct that this limits the scope of the data. To address this concern, we have included another 11 months of sensor data (1 full week from each month), 24 hours per day including weekends. We hope this will be useful in learning a wider range of traffic conditions. Parsing the incident logs and verifying anomalies is an intensive task, so we will continue to work on this so that future researchers may validate their performance on this expanded dataset.
>
> >The paper acknowledges challenges with labeling anomalies due to delays and potential inaccuracies in manual incident reporting. The methods used for labeling and evaluating the models, such as assuming a fixed delay for incident reporting, may not accurately reflect real-world conditions. It would be better to discuss further approaches to address the issues caused by delays.
>
> **Response:** To clarify, we did not assume a fixed delay for incident reporting. There is some unknown delay in incident reporting, which makes model evaluation difficult for this task. As a result, we assumed a maximum delay in incident reporting. If a model detected an anomaly during this window, we considered the prediction a true positive. Extending the time for a true positive before or after an incident was reported has been done in recent literature as well [1, 2]. Further research needs to be done in the domain to determine what bounds are reasonable. Methods to address this problem include: consulting domain experts to determine when a crash happened, creating new evaluation metrics that are less sensitive to the maximum delay threshold, or performing a field test to assess the timeliness of the anomaly predictions. We will add this discussion to the limitations/conclusion section of the paper to guide future research in this area.
>
> [1] LIYANAGE, Y. W., ZOIS, D.-S., AND CHELMIS, C. Near real-time freeway accident detection. IEEE transactions on intelligent transportation systems 23, 2 (2020), 1467–1478.
>
> [2] DENG, L., LIAN, D., HUANG, Z., AND CHEN, E. Graph convolutional adversarial networks for spatiotemporal anomaly detection. IEEE Transactions on Neural Networks and Learning Systems 33, 6 (2022), 2416–2428.

---

> > ### Comment · Area_Chair_tVE6 · 2024-08-29
> >
> > Can Reviewer 9FVR please comment on the authors' rebuttal? Thank you.

---

> > ### Comment · Reviewer_9FVR · 2024-08-29
> > **Thanks for the response!**
> >
> > I have read the authors' responses and other reviews. My questions have been addressed. Thus, I will maintain my positive score.

---

### Author Response · Authors · 2024-08-16
**Summary of New Data and Baselines**

We want to thank all the reviewers for their detailed feedback! In this comment, we discuss some steps we have taken to address common concerns with the paper.

### New Data
All this data can be accessed under the Supplementary Data section of the README in the data repository (https://github.com/acoursey3/freeway-anomaly-data).

1. We added 24-hour sensor data for 1 full week from 11 months from July 2023 to June 2024. We have processed the data in the same way the data from October that comprises the main dataset. We hope this will address concerns with seasonality, as models can be learned or applied to data from the full seasonal spectrum.

2. We added raw event logs for all data. These event logs contain additional contextual information, such as the type of incident that was reported.

3. We will manually annotate the timestamp when we expect a reported crash to have actually occurred by inspecting the time-space diagram. This could be used to improve the conservatism of the definition of a true positive. However, note that measuring and reducing the reduction in reporting delay should still be the main task of the dataset.

### New Baselines
1. Non-reconstruction-based baselines. We obtained reasonable performance with a clustering approach and observed a rule-based approach led to a high false positive rate. See the results below.

The first additional approach we evaluated was K Means clustering. We increased the number of clusters on the anomaly-free training data until the inertia (within-cluster sum-of-squares) converged. This led to 3 clusters. Then, we determined the distance of each test point from the cluster centroids and varied the distance threshold for anomaly detection. As in the main results, we fixed this threshold to demonstrate the performance with a 5\% false positive tolerance.

Next, we implemented a widely used rule-based algorithm, the California algorithm [1]. This simple technique determines if a node is in an anomalous state by observing the upstream and downstream occupancy. It uses predefined thresholds that are known to be hard to tune. It typically "requires the laborious calculation of thresholds for each location where it is installed. In large networks, separate thresholds must be calculated for different road geometries'' [2]. As a starting point, we used thresholds from the original paper [1] (T1=8.1, T2=0.313, T3=16.8). Then, we attempted to optimize the three thresholds using a Python optimization library to fit the 5\% false positive tolerance. The optimizer maintained the original thresholds. This optimization is more difficult than for the other methods, as three variables need to be optimized at once, and the evaluation of the algorithm is computationally expensive. These thresholds led to about 8\% of the nodes being false positives. However, it detected an anomalous node 99\% of the time. Consequentially, it never failed to detect the time a crash was reported. Practically, this means the traffic management center would constantly be alerted that an incident exists.

The results of both experiments are shown in the table below. The clustering approach leads to reasonable performance, detecting half the crashes before they are reported on average. It can maintain a 5\% false positive rate. The California algorithm perfectly detects crashes due to its high false positive rate. This highlights the benefits of the other approaches proposed in the paper with thresholds that can easily be tuned. In both cases, more advanced approaches should be benchmarked in future work.

| Model       | Reporting Delay  | Miss Percentage | FPR  | AUC  |
|-------------|------------------|-----------------|------|------|
| K Means     | -1.58 ± 10.41    | 50%             | 5%   | 0.60 |
| California  | -15 ± 0          | 0%              | 99%  | N/A  |

[1] Payne, Howard J., and Samuel C. Tignor. "Freeway incident-detection algorithms based on decision trees with states." Transportation Research Record 682 (1978).

[2] MARTIN, P. T., PERRIN, J., HANSEN, B., KUMP, R., MOORE, D., ET AL. Incident detection algorithm evaluation. Tech. rep., Upper Great Plains Transportation Institute, 2001.

---

### Decision · Program_Chairs · 2024-09-26

**Decision:**

Accept (Poster)

**Comment:**

This paper received 5 positive reviews, with the average rating of 6.67. The average rating places this paper below the acceptance threshold.